# The GET insertase exhibits conformational plasticity and induces membrane thinning

Melanie A. McDowell [1,4,5] ✉, Michael Heimes [1,5], Giray Enkavi [2], Ákos Farkas[3], Daniel Saar [1], Klemens Wild [1], Blanche Schwappach [3], Ilpo Vattulainen [2] & Irmgard Sinning [1] ✉

The eukaryotic guided entry of tail-anchored proteins (GET) pathway mediates the biogenesis of tail-anchored (TA) membrane proteins at the endoplasmic reticulum. In the cytosol, the Get3 chaperone captures the TA protein substrate and delivers it to the Get1/Get2 membrane protein complex (GET insertase), which then inserts the substrate via a membrane-embedded hydrophilic groove. Here, we present structures, atomistic simulations and functional data of human and *Chaetomium thermophilum* Get1/Get2/Get3. The core fold of the GET insertase is conserved throughout eukaryotes, whilst thinning of the lipid bilayer occurs in the vicinity of the hydrophilic groove to presumably lower the energetic barrier of membrane insertion. We show that the gating interaction between Get2 helix α3' and Get3 drives conformational changes in both Get3 and the Get1/Get2 membrane heterotetramer. Thus, we provide a framework to understand the conformational plasticity of the GET insertase and how it remodels its membrane environment to promote substrate insertion.

Around 5% of eukaryotic membrane proteins are anchored in the lipid bilayer by a single transmembrane domain (TMD) at their extreme C-terminus[1]. These so-called tail-anchored (TA) proteins reside in almost every cellular membrane and are vital to processes such as vesicular trafficking, protein translocation and apoptosis[2]. During their biogenesis, TA proteins destined for membranes of the secretory pathway must be recognised post-translationally in the cytosol, targeted to the endoplasmic reticulum (ER) and inserted into the membrane[3]. One dedicated route for TA protein targeting and insertion at the ER is the extensively studied guided-entry of TA proteins (GET) pathway[4], which is conserved throughout eukaryotes and operates in parallel to the more recently characterised ER membrane protein complex (EMC) and signal recognition particle independent (SND) pathways[5,6].

Within the membrane targeting stages of the GET pathway, the homodimeric ATPase Get3 (also called TRC40 in metazoans) adopts closed and open conformations in response to nucleotide load and interactions with other pathway components, allowing binding and release of the TA protein respectively[7–12]. In the ADP-P$_i$ form, closed Get3 shields the substrate TMD within a hydrophobic groove to protect it from the aqueous cytosol[13]. This complex is then captured at the ER membrane by the cytoplasmic domains (CDs) of the GET insertase, a membrane protein complex comprising Get1/Get2[14–17] (also called WRB/CAML in metazoans). The flexible, N-terminal Get2-CD first binds the Get3/TA protein complex, allowing the Get1-CD coiled-coil to insert at the Get3 dimer interface[18,19]. These interactions result in opening of the Get3 dimer, nucleotide dissociation and release of the TA protein to the ER membrane through disruption of the hydrophobic groove[11,15,18–20].

The TA protein then engages with the TMDs of the GET insertase[21], the minimal machinery required for membrane insertion[15,17,22,23]. Our understanding of the mechanism of TA protein insertion was recently

[1]Heidelberg University Biochemistry Center (BZH), Im Neuenheimer Feld 328, 69120 Heidelberg, Germany. [2]Department of Physics, University of Helsinki, P. O. Box 64, FI-00014 Helsinki, Finland. [3]Department of Molecular Biology, University Medical Center Göttingen, 37073 Göttingen, Germany. [4]Present address: Max Planck Institute of Biophysics, Max-von-Laue Strasse 3, 60438 Frankfurt am Main, Germany. [5]These authors contributed equally: Melanie A. McDowell, Michael Heimes. ✉e-mail: melanie.mcdowell@biophys.mpg.de; irmi.sinning@bzh.uni-heidelberg.de

advanced by a cryo-electron microscopy (cryo-EM) structure of human Get1/Get2 (*hs*Get1/Get2) bound to *hs*Get3 via the *hs*Get1-CD[24]. The *hs*Get1 TMDs form a hydrophilic groove open to the cytosol[24], which is conserved amongst the insertases of the Oxa1 superfamily, including bacterial YidC, the Emc3 subunit of the EMC and TMCO1 within the multipass translocon[25,26]. The hydrophilic groove directly contacts the TA protein[21] and is likely to facilitate insertion by providing a transient binding site for the substrate's polar C-terminal extension (CTE) as it traverses the ER membrane. In addition, a previously unidentified CD within *hs*Get2 (helix α3') interacts hydrophobically with the TA protein binding domain (TABD) of *hs*Get3, which was itself rearranged relative to other Get3 structures[24]. Given that this gating interaction is proximal to the disrupted substrate binding site and was shown to be functionally important in *hs*Get2 and *Saccharomyces cerevisiae* Get2 (*sc*Get2), it was proposed that helix α3', a shorter mimic for the substrate TMD, actively drives TA protein insertion[24]. However, the precise molecular function of helix α3' is yet to be fully defined. The hydrophilic groove and helix α3' are self-contained within a *hs*Get1/Get2 heterodimer, the minimal unit shown to be sufficient for substrate insertion in vitro[23]. However, the cryo-EM structures and native mass spectrometry of *hs*Get1/Get2 and *sc*Get1/Get2 show they form a 2:2 heterotetramer, which is stabilised by Get3 and interfacial lipid binding and is important for efficient TA protein insertion[24].

Although *S. cerevisiae* has long been the model system to study the GET pathway, a reconstruction of the yeast GET insertase was of insufficient resolution to determine the TMD arrangement[24], therefore the extent to which the structure of Get1/Get2 is conserved is unclear. Similarly, given that Get1/Get2 is stabilised by amphipols in existing reconstructions, it is not known how the arrangement of the TMDs is influenced by the lipid bilayer and vice versa. Furthermore, although the conformational landscape of Get3 is well characterised, the plasticity of its complex with the GET insertase has not been investigated.

Here, we show that the Get1/Get2 heterotetramer and Get3 TABD undergo conformational changes in response to the gating interaction between helix α3' and Get3. We determine the structure of the GET insertase from *Chaetomium thermophilum* and resolve additional features in the human insertase, showing that the overall structure of the Get1/Get2 heterodimer is conserved throughout eukaryotes. In addition, both complexes remodel the membrane, as thinning of the lipid bilayer in the vicinity of the hydrophilic groove is observed, akin to other membrane insertases. These molecular details of the GET insertase extend a mechanistic model for TA protein insertion.

## Results

### Helix α3' dictates the conformation of the *hs*Get1/Get2 heterotetramer

In our recent structure of the human GET insertase, we identified an unexpected structural element (helix α3') in *hs*Get2 that contacts the *hs*Get3 TABD, and showed that this element impacts on TA protein insertion[24]. To understand the molecular function of helix α3' in detail, we now investigated the effect of a polyglycine substitution of helix α3' (Δα3')[24] on the structure of the human GET insertase. As previously for the sequence with native helix α3' (referred to henceforth as 'wild type')[24], a heterodimeric fusion with the flexible N-terminus of Get2 truncated (*hs*Get2^ΔN/Δα3'-Get1) was purified and complexed with apo *hs*Get3. The complex was subsequently reconstituted in PMAL-C8 amphipol and a single-particle cryo-EM reconstruction was refined to 4.2 Å resolution (Supplementary Figs. 1 and 2 and Table 1). Interestingly, the membrane-embedded region in the Δα3' structure shows pronounced differences compared to the wild type complex, with density for only seven of the twelve TMDs in the *hs*Get1/Get2 heterotetramer (Fig. 1A). Rigid body docking of the *hs*Get1/Get2 model into this density revealed that only one *hs*Get1/Get2^Δα3' heterodimer half is resolved (Fig. 1B) and adopts an identical TMD arrangement (Supplementary Fig. 3A). The remaining TMD is contiguous with the second

coiled-coil and thus corresponds to *hs*Get1 TMD2 from the other heterodimer half (Fig. 1B).

Notably, the structure reveals that the Δα3' variant causes a complete repositioning of the resolved *hs*Get1/Get2^Δα3' heterodimer within the complex (Supplementary Fig. 3B). Superimposition of *hs*Get3 between the wild type and Δα3' complexes highlights a twist in their respective interactions with the *hs*Get1-CD, with the coiled-coils being rotated and tilted by ~15° between structures (Fig. 1C). This relatively small change in the Get1/Get3 interaction is propagated via the coiled-coils in a lever-like manner, inevitably having a large impact on the structure of the membrane heterotetramer. A model for the missing *hs*Get1/Get2^Δα3' TMDs based on superimposition with the resolved *hs*Get1 coiled-coil suggests that the membrane heterotetramer has a converse orientation to the wild type GET insertase: the hydrophilic groove is rotated into the heterotetramer interface, whilst *hs*Get2 TMD1/TMD2 are moved to the periphery of the complex (Fig. 1D). In addition, the N-terminus of *hs*Get2 TMD3, and thus the helix α3' substitution, are no longer positioned proximal to the *hs*Get3 TABD as in the wild type structure. Instead, the N-terminus of *hs*Get2^Δα3' TMD3 abuts *hs*Get1 TMD2 of the opposite heterodimer, burying a surface area of 190 Å² (Fig. 1D). Therefore, our data suggest that the *hs*Get1/Get2 membrane heterotetramer can adopt different conformations, with helix α3' interactions dictating the preferred arrangement.

### The core structure of the Get1/Get2 heterodimer is conserved across eukaryotes

To further investigate the conformation of the GET insertase from another organism, we took advantage of the thermostability of Get1, Get2 and Get3 proteins from the thermophilic fungus *Chaetomium thermophilum*. A heterodimeric fusion with the flexible N-terminus of Get2 truncated (*ct*Get2^ΔN-Get1) was recombinantly produced from *S. cerevisiae* and complexed with apo *ct*Get3 purified from *Escherichia coli*. We subsequently obtained single-particle cryo-EM reconstructions of both amphipol- and nanodisc-reconstituted *ct*Get2^ΔN-Get1/Get3 at an overall resolution of 5.0 Å and 4.6 Å respectively (Supplementary Figs. 1 and 4 and Table 1). The overall structure of the GET insertase is the same in both environments (Fig. 2A and Supplementary Fig. 5A), showing that two *ct*Get1-CDs are associated with the open conformation of *ct*Get3 and thus that a *ct*Get1/Get2 symmetric heterotetramer is present in the membrane, as previously observed for the wild type *S. cerevisiae* and human insertases[24]. In both reconstructions, there is clear density in the membrane region for two three-helix bundles at a local resolution of 6.5 Å. However, the reconstruction in amphipol additionally revealed two ~60 Å bridging helices tilted at 75° relative to the membrane normal that brace these three-TMD halves on either side (Fig. 2A, red helix). This density map was therefore more complete and was used to build an initial model. Superimposition of the three-helix bundle with the structure of *hs*Get1/Get2[24] showed a conserved arrangement with *hs*Get1, allowing their assignment as *ct*Get1 TMDs (Fig. 2B and C). Accordingly, *ct*Get1 creates a hydrophilic groove in the membrane, lined by hydrophilic or charged residues like N18, R110, T114, R115, Q118 and E174 (Supplementary Fig. 5B). *ct*Get1 TMD1/TMD2 are also contiguous with the density for the coiled-coil (Supplementary Fig. 5C); as in *hs*Get1, TMD2 forms a long continuous helix with the CD, whilst an amphipathic helix (AH) connects TMD1 at the cytoplasmic membrane interface (Supplementary Fig. 5D). Notably, our comparison revealed the long bridging helices to be *ct*Get2 TMD3, which adopt the same conformation relative to *ct*Get1 as in *hs*Get1/Get2 (Fig. 2B and C). As observed for *hs*Get1 W158 (Supplementary Fig. 5E), aromatic stacking by the equivalent W161 in *ct*Get1 TMD3 is likely to contribute to the interaction with *ct*Get2 TMD3 (Supplementary Fig. 5F).

The ER cap, the loop connecting TMD2 and TMD3 of Get1 homologues, was previously observed to make intimate contacts with

**Table 1 | Cryo-EM data collection, refinement and validation statistics**

| | H. sapiens Get2$^{\Delta N/\Delta\alpha3'}$-Get1/ Get3 in PMAL-C8 (EMDB-16802) (PDB 8CR2) | H. sapiens Get2$^{\Delta N}$-Get1/ Get3 in PMAL-C8 (EMDB-16801) (PDB 8CR1) | C. thermophilum Get2$^{\Delta N}$-Get1/ Get3 in A835 (EMDB-16817) (PDB 8ODU) | C. thermophilum Get2$^{\Delta N}$-Get1/ Get3 in nanodisc (EMDB-16819) (PDB 8ODV) |
|---|---|---|---|---|
| **Data collection and processing** | | | | |
| Detector | K3 | K2 | K3 | K3 |
| Magnification | 81,000 | 165,000 | 64,000 | 81,000 |
| Voltage (kV) | 300 | 300 | 300 | 300 |
| Electron exposure (e–/Å$^2$) | 53.2 | 46 | 55 | 60.8 |
| Defocus range (μm) | 1.2-2.4 | 0.8–2.0 | 1.5–3.5 | 1.2–2.4 |
| Pixel size (Å) | 1.11 | 0.81 | 1.375 | 1.11 |
| Symmetry imposed | C1 | C2 | C1 | C2 |
| Initial particle images (no.) | 1,995,680 | 1,561,837 | 4,337,300 | 5,218,800 |
| Final particle images (no.) | 224,354 | 189,844 | 796,684 | 259,692 |
| Map resolution (Å) | 4.2 | 3.2 | 5.0 | 4.7 |
| FSC threshold | 0.143 | 0.143 | 0.143 | 0.143 |
| Map resolution range (Å) | 4.0-7.7 | 3.1–5.4 | 4.7–9.0 | 4.1–8.5 |
| **Refinement** | | | | |
| Initial model used (PDB code) | 6SO5 | 6SO5 | 3IQW, 3SJA | (this paper) |
| Model resolution (Å) | 4.0 | 3.2 | 4.8 | 4.5 |
| FSC threshold | 0.143 | 0.143 | 0.143 | 0.143 |
| Map sharpening B factor (Å$^2$) | -202 | -58 | -512 | -285 |
| Model composition | | | | |
| Non-hydrogen atoms | 6838 | 9205 | 8420 | 8233 |
| Protein residues | 855 | 1158 | 1044 | 1022 |
| Ligands | ZN:1 | ZN:1 | ZN:1 | ZN:1 |
| B factors (Å$^2$) | | | | |
| Protein | 65 | 121 | 191 | 212 |
| Ligand | 97 | 189 | 186 | 195 |
| R.m.s. deviations | | | | |
| Bond lengths (Å) | 0.004 | 0.003 | 0.003 | 0.007 |
| Bond angles (°) | 1.075 | 0.726 | 0.793 | 1.098 |
| Validation | | | | |
| MolProbity score | 2.18 | 1.73 | 1.91 | 2.22 |
| Clashscore | 19.83 | 10.58 | 14.05 | 25.51 |
| Poor rotamers (%) | 0.00 | 0.00 | 0.00 | 0.00 |
| Ramachandran plot | | | | |
| Favoured (%) | 94.24 | 96.85 | 96.18 | 95.39 |
| Allowed (%) | 5.76 | 3.15 | 3.82 | 4.61 |
| Disallowed (%) | 0.00 | 0.00 | 0.00 | 0.00 |

hsGet1 TMD1 and hsGet2 TMD2/TMD3[24] and thus is likely to be intrinsic to the fold and stability of the heterodimer. However, the resolution of the human GET insertase reconstruction precluded building of an ab initio model for the ER cap[24]. Re-refinement of the original data using CryoSPARC[27,28] resulted in a significant improvement in the overall resolution of hsGet2$^{\Delta N}$-Get1/Get3 to 3.2 Å (Fig. 2D, Supplementary Fig. 6A, B and Table 1). Although the resolution of the membrane region was still lower (4.5–5.0 Å; Supplementary Fig. 6C), density for both the intra- and intermolecular disulphide bonds and for bulky side chains within the TMDs were more clearly defined (Supplementary Fig. 5E), confirming the register of the original model. In addition, a model for the ER cap could be derived from an AlphaFold[29] model of hsGet1/Get2 and docked precisely into the corresponding density,

allowing it to be incorporated into our structure (Fig. 2D and Supplementary Fig. 7A). AlphaFold[29] was similarly used to build the ER cap of ctGet1 in our models for both the amphipol and nanodisc reconstructions (Fig. 2A, Supplementary Figs. 5A and 7B). The overall fold of the ER cap is similar between ctGet1 and hsGet1 (Fig. 2E) and is characterised by four highly conserved proline residues (Supplementary Fig. 8A) that demarcate kinks in the protein backbone (Supplementary Fig. 8B, C). Our model shows that the ER cap stabilises the heterodimer interface largely through hydrophobic interactions with the luminal ends of hsGet2 TMD2/TMD3, including aromatic stacking with a conserved tryptophan (W135 in hsGet1 and W137 in ctGet1) in the cap (Supplementary Fig. 8D).

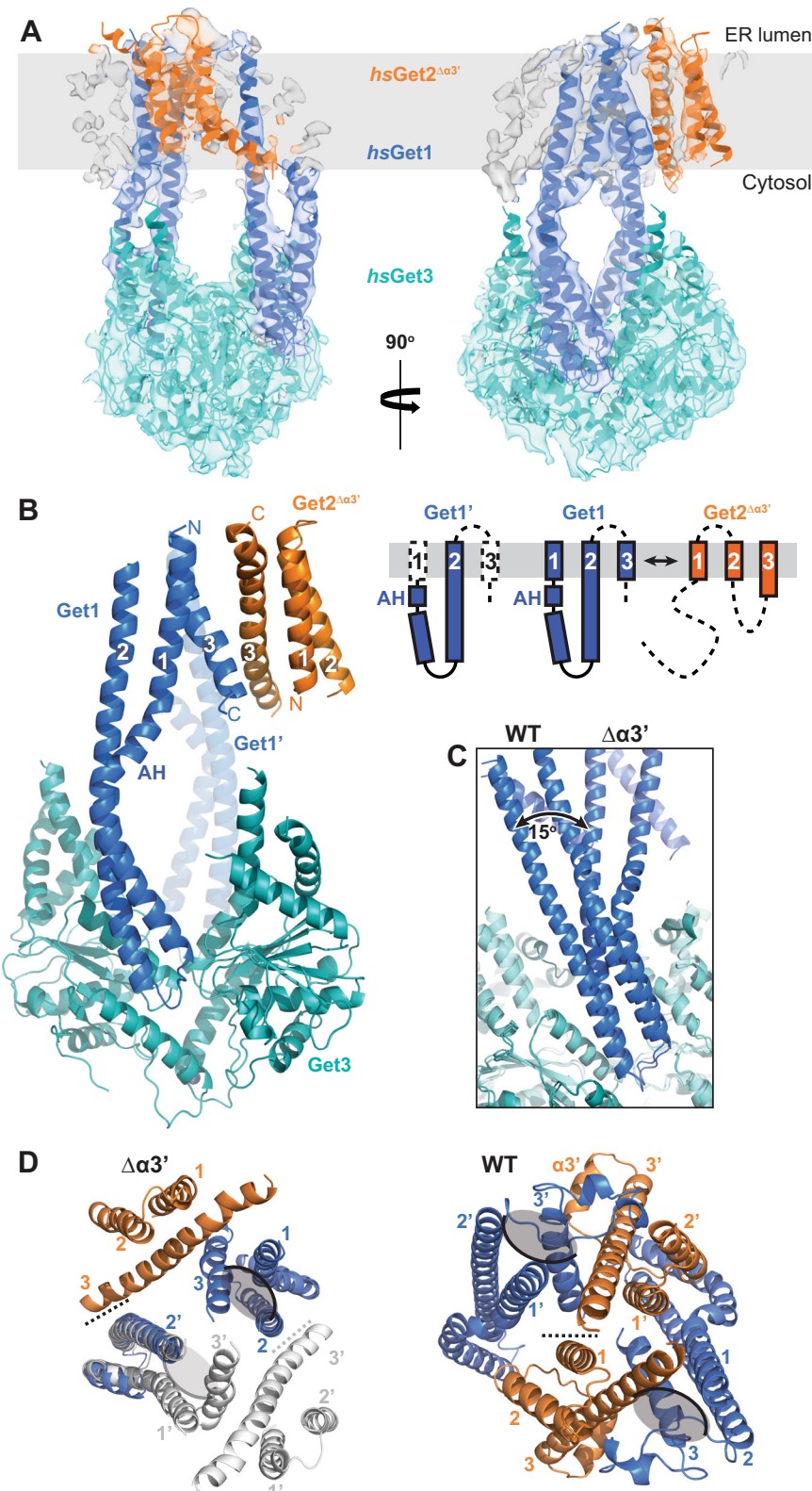

**Fig. 1 | Deletion of helix α3′ leads to rearrangement of the Get1/Get2 hetero-tetramer. A** Model for *hs*Get2^ΔN/Δα3′-Get1/Get3 in PMAL-C8 amphipol superposed to the cryo-EM density. Membrane plane defined perpendicular to the Get3 symmetry axis with boundaries inferred from the location of the TMDs. **B** Structure of *hs*Get2^ΔN/Δα3′-Get1/Get3. The schematic shows the topology of each copy of *hs*Get1 and *hs*Get2 in the structure, with dashed lines representing features not present in our model. **C** Superimposition of the wild type (WT) and Δα3′ *hs*Get1/Get2/Get3 complexes via *hs*Get3 (RMSD 1.71 Å over 503 Cα atoms), showing differences in the relative tilt of *hs*Get1-CD. **D** View of the Δα3′ and WT *hs*Get1/Get2 heterotetramers from the ER lumen after superimposition as in (**C**). The partially resolved *hs*Get1/Get2^Δα3′ heterodimer (light grey) is modelled by superimposition of *hs*Get1 from the resolved heterodimer (RMSD 1.89 Å over 96 Cα atoms). The heterotetramer interface is represented by a dashed line and the hydrophilic grooves are indicated by grey ovals.

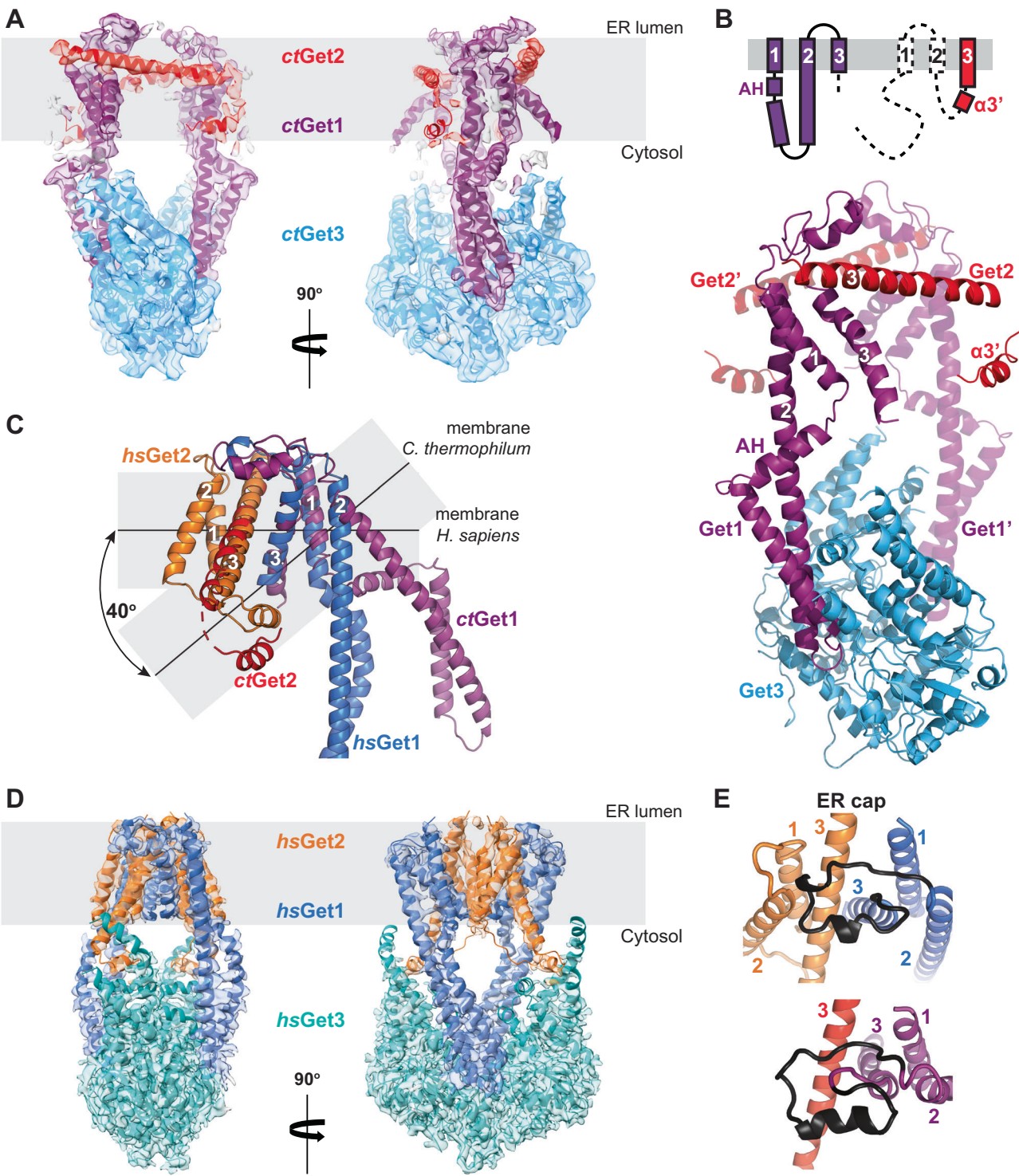

**Fig. 2 | The structure of the Get1/Get2 heterodimer is conserved between *H. sapiens* and *C. thermophilum*. A** Model for *ct*Get2^ΔN-Get1/Get3 in A835 amphipol superposed to the cryo-EM density. Membrane plane defined perpendicular to the Get3 symmetry axis with boundaries inferred from the location of the TMDs. **B** Structure of *ct*Get2^ΔN-Get1/Get3. The schematic shows the topology of *ct*Get1 and *ct*Get2, with dashed lines representing features not present in our model. **C** Superimposition of *ct*Get2^ΔN-Get1 TMDs with *hs*Get2^ΔN-Get1 (*hs*Get1 TMDs 1-3

and *hs*Get2 TMD3; RMSD 2.69 Å over 79 Cα atoms). The membrane planes are defined as in Figures (**A**–**D**). **D** Model for *hs*Get2^ΔN-Get1/Get3 in PMAL-C8 amphipol superposed to the cryo-EM density reprocessed from EMDB-10266[24]. Membrane plane defined perpendicular to the Get3 symmetry axis with boundaries inferred from the location of the TMDs. **E** Side-by-side comparison of the ER cap (black) from *hs*Get1 (residues 126-150) and *ct*Get1 (residues 128-153) after superimposition as shown in (**C**).

Weak density also shows that helix α3' is present in the same conformation relative to *ct*Get2 TMD3 in both the amphipol and nanodisc reconstruction (Supplementary Fig. 8E, F). Furthermore, the density map for the nanodisc-reconstituted sample provides evidence that *ct*Get2 TMD1 and TMD2 occupy the same position in

the heterodimer as the equivalent *hs*Get2 TMDs (Supplementary Fig. 8G), albeit they are poorly resolved and likely exhibit flexibility on the edge of the complex. Taken together, we observe that overall the core fold of the Get1/Get2 heterodimer is conserved from lower to higher eukaryotes.

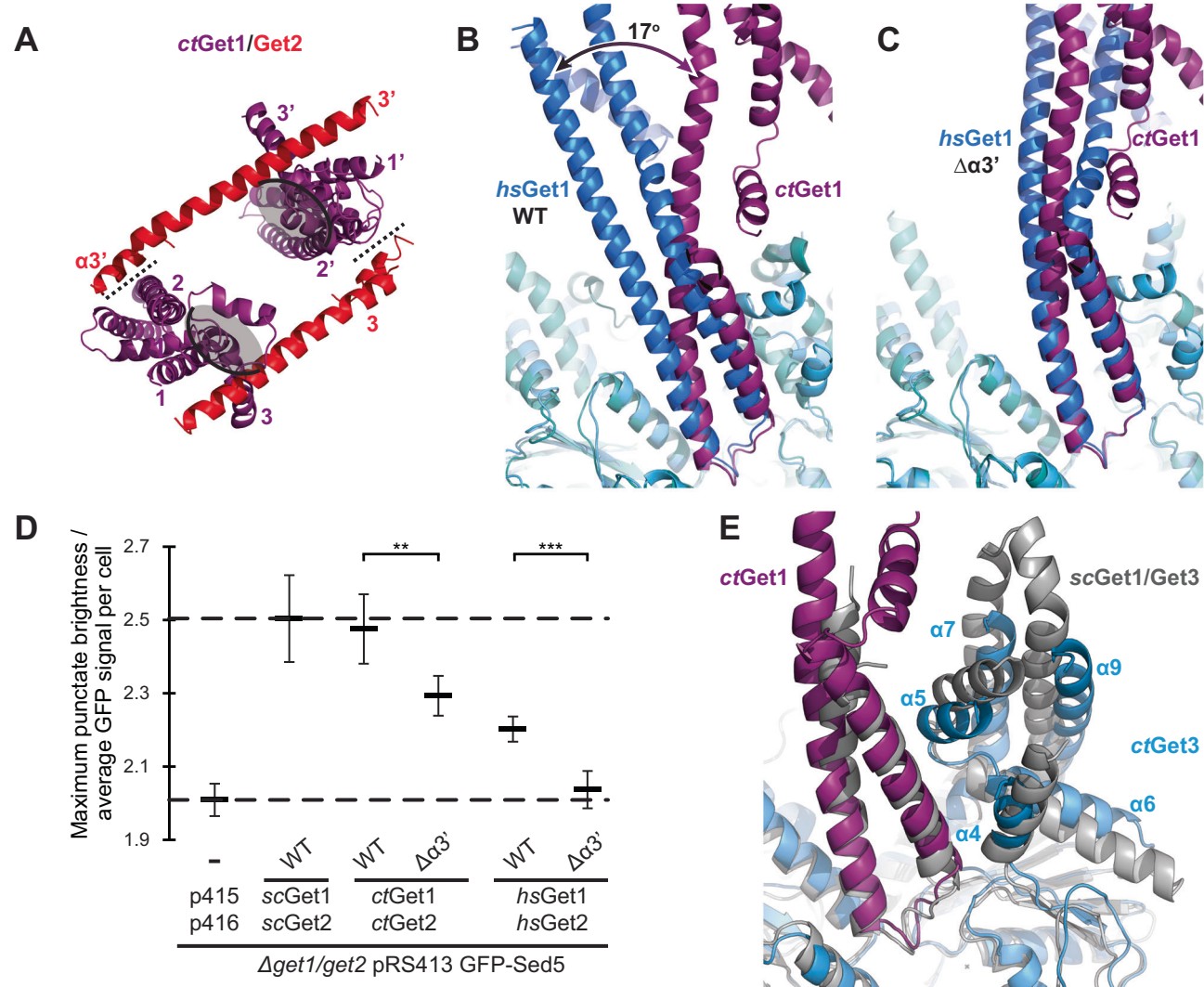

**Fig. 3 | The *H. sapiens* and *C. thermophilum* Get1/Get2 heterotetramers adopt different conformations. A** View of the amphipol *ct*Get1/Get2 heterotetramer from the ER lumen after superimposition via Get3 with the wild type (WT) *hs*Get2$^{\Delta N}$-Get1/Get3 complex as positioned in Fig. 1D (RMSD 1.71 Å over 503 Cα atoms). The heterotetramer interface is represented by a dashed line and the hydrophilic grooves are indicated by grey ovals. **B** Superimposition of the *ct*Get2$^{\Delta N}$-Get1/Get3 and *hs*Get2$^{\Delta N}$-Get1/Get3 complexes as shown in **A** (RMSD 2.27 Å over 475 Cα atoms), showing the difference in the relative tilt of the Get1-CD. **C** Superimposition of the amphipol *ct*Get2$^{\Delta N}$-Get1/Get3 and *hs*Get2$^{\Delta N/\Delta\alpha3'}$-Get1/Get3 complexes via Get3 (RMSD 1.76 Å over 484 Cα atoms), showing similarity in the position of the Get1-CD. **D** *sc*Get1-4PC/Get2-4PC, *ct*Get1/Get2 or *hs*Get1/Get2 were co-expressed from the indicated plasmids in *Δget1/get2* yeast strains for WT and Get2$^{\Delta\alpha3'}$ sequences. Transformants with the empty vector were taken as a negative control (-).

Quantification of GFP-Sed5 distribution from fluorescence microscopy images of *Δget1/get2* yeast cells expressing GFP-Sed5 together with *sc*Get1/Get2, *ct*Get1/Get2 or *hs*Get1/Get2 constructs. The dashed line at the bottom shows a lower maximum punctate fluorescence relative to average cellular GFP fluorescence for cells carrying empty plasmid, whilst the dashed line at the top shows a higher ratio for cells expressing WT *sc*Get1/Get2. Several hundreds of cells were quantified for each replicate and error bars indicate standard error calculated from six independent experiments. Statistically significant differences between the WT and Get2$^{\Delta\alpha3'}$ variants were determined using the two-sided Welch's *t*-test: **: 0.0086, ***: 0.0005. See also Supplementary Fig. 9. **E** Superimposition of the *ct*Get3 amphipol dimer (blue) with open *sc*Get3 (grey; RMSD 2.50 Å over 534 Cα atoms; PDB accession 3SJA).

## The *ct*Get1/Get2 heterotetramer has a different conformation from *hs*Get1/Get2

Despite the conserved core fold, the overall morphology of the *ct*Get1/Get2 heterotetramer is strikingly different from the wild type human structure, as a large central membrane cavity separates the heterodimer halves (Fig. 2A). In addition, the hydrophilic groove from each heterodimer half points towards the central membrane cavity rather than to the outside, as observed in the *hs*Get1/Get2 heterotetramer (Figs. 3A and 1D). Conversely, *ct*Get2 TMD1 and TMD2 are placed at the periphery of the complex (Supplementary Fig. 8G), in contrast to the equivalent TMDs of *hs*Get2 being intimately arranged at the heterotetramer interface. Finally, *ct*Get2 helix α3' and the adjoining TMD3 N-terminus form the sole contact between *ct*Get1/Get2 heterodimers

with an interface of only ~120 Å², binding the opposing *ct*Get1 TMD2 instead of the *ct*Get3 TABD. Superimposition of *ct*Get3 with *hs*Get3 demonstrates that these changes are caused by a ~17° twist and tilt of the *ct*Get1 coiled-coil (Fig. 3B). Remarkably, this position of the *ct*Get1 CD and the resulting orientation of *ct*Get1/Get2 in the heterotetramer are largely similar to the conformation observed in the *hs*Get1/Get2$^{\Delta\alpha3'}$ variant (Figs. 3A, C and 1D). However, the model for the unresolved second *hs*Get1/Get2$^{\Delta\alpha3'}$ heterodimer suggests that the central membrane cavity is expected to be much smaller than between the *ct*Get1/Get2 heterodimer halves (Fig. 3A and 1D). Indeed, the *ct*Get1 coiled-coil is rotated even further outwards compared to *hs*Get1 (Fig. 3C), resulting in a greater separation of subunits. Nevertheless, the structures of the wild type *C. thermophilum* and Δα3' *H. sapiens* GET

insertases represent analogous, alternative conformations of a largely conserved membrane heterotetramer, which we define as state 1. Accordingly, the conformation of the wild type *hs*Get1/Get2 heterotetramer, where helix α3' is bound to Get3, is defined as state 2.

Given that helix α3' occupies distinctly different binding sites within the *C. thermophilum* and *H. sapiens* GET insertase, we probed whether this element is functionally important in both sequences. In line with previous experiments conducted with mouse Get2[24], the Δα3' variant of *hs*Get2 does not restore the growth phenotype of the *S. cerevisiae* Δget1/get2 strain when coexpressed with *hs*Get1 (Supplementary Fig. 9A). In contrast, the analogous *ct*Get1/Get2^Δα3' variant appears to fully complement the Δget1/get2 strain. Despite impaired expression of *hs*Get1/Get2^Δα3' (Supplementary Fig. 9B), both mutant insertases can still recruit GFP-tagged Get3 to the ER membrane (Supplementary Fig. 9C), implicating correct membrane integration. Crucially, however, the GFP-tagged TA protein Sed5 is misolocalised in the presence of both *ct*Get1/*ct*Get2^Δα3' and *hs*Get1/Get2^Δα3' relative to the respective wild type complexes (Fig. 3D and Supplementary Fig. 9D). Therefore, helix α3' appears to be functionally important throughout eukaryotes and could occupy different binding sites within the GET insertase as part of this role.

## Helix α3' binding also correlates with rearrangement of the *hs*Get3 TABD

In addition to these conformational changes in the membrane heterotetramer, we previously observed that helices α4-α9 of the *hs*Get3 TABD show large rearrangements when bound to *hs*Get2 helix α3' in state 2[24]. In contrast, *ct*Get3 is not interacting with *ct*Get2 helix α3' within our cryo-EM reconstructions and shows no structural rearrangements with respect to the majority of high-resolution Get3 structures (Fig. 3E and Supplementary Data 1). Therefore, we questioned whether this alternative conformation is specific to *hs*Get3 from higher eukaryotes or correlated with the interaction of helix α3'. As the Get3 TABD helices α4 and α5 are not resolved in the *hs*Get2^ΔN/Δα3'-Get1/Get3 cryo-EM structure, we bound *E. coli* purified *hs*Get3 to the *hs*Get1-CD in the absence of nucleotide (Supplementary Fig. 10A) and solved the crystal structure of this complex to 2.8 Å resolution (Table 2). Surprisingly, the structure showed a heterodimer (Fig. 4), despite multi-angle light scattering (MALS) confirming the *hs*Get3/Get1-CD complex was present in solution as the expected heterotetramer (Supplementary Fig. 10B). This indicates that dissociation of the *hs*Get3 homodimer occurred during crystallisation, as previously observed for *sc*Get3[9]. The interfacial zinc ion is accordingly absent in the electron density, asserting that zinc coordination is important for dimerisation. There are otherwise only minor rearrangements of structural elements found at the dimer interface within the *hs*Get2^ΔN-Get1/Get3 cryo-EM structures (Fig. 4A), indicating the *hs*Get3 fold is independent of oligomeric state. In the crystal structure, the *hs*Get1-CD contacts the *hs*Get3 monomer via the more extensive interface I rather than the smaller interface II (classified previously for *sc*Get3/Get1-CD complexes[18]). Although the coiled-coil adopts a tilt/twist most similar to the wild type human GET insertase cryo-EM structure (Supplementary Fig. 10C), it is difficult to make a biological interpretation of this position in the absence of interface II with *hs*Get3 and the adjoining TMDs.

In the *hs*Get3/Get1-CD crystal structure, helices α4 and α5 of the *hs*Get3 TABD are unambiguously in the same conformation as in the cryo-EM structure of the *ct*Get1/Get2/Get3 complex (as in state 1), rather than the alternative conformation present within the *hs*Get2^ΔN-Get1/Get3 structure where α4 and α5 are reorientated by ~90° and 180° respectively (as in state 2) (Fig. 4B). Therefore, the *hs*Get3 TABD can adopt either state 1 or state 2, with the latter correlated with interactions with helix α3' and/or the *hs*Get1-CD via interface II. To gain further insights into these different TABD conformations, we compared our structures with all other

## Table 2 | Crystallographic data collection and refinement statistics (molecular replacement)

|  | *H. sapiens* Get3/Get1-CD PDB 8CQZ |
| --- | --- |
| **Data collection** |  |
| Space group | P2₁2₁2₁ |
| Cell dimensions |  |
| a, b, c (Å) | 69.48, 81.55, 92.19 |
| α, β, γ (°) | 90.00, 90.00, 90.00 |
| Resolution (Å) | 45.9-2.8 (2.9-2.8)* |
| $R_{pim}$ | 0.018 (0.448) |
| I / σI | 25.0 (1.4) |
| Completeness (%) | 100 (100) |
| Redundancy | 12.8 (13.3) |
| **Refinement** |  |
| No. reflections | 12947 (1107) |
| $R_{work}$ / $R_{free}$ | 0.24 / 0.28 |
| No. atoms |  |
| Protein | 3169 |
| Ligand/ion | 0 |
| Water | 0 |
| Average B-factor (Å²) | 151 |
| R.m.s. deviations |  |
| Bond lengths (Å) | 0.002 |
| Bond angles (°) | 0.480 |
| Ramachandran plot |  |
| Favoured (%) | 96.71 |
| Allowed (%) | 3.29 |
| Disallowed (%) | 0.00 |

*Values in parentheses are for highest-resolution shell.

deposited structures of Get3 alone and in complex with its different binding partners (Supplementary Data 1). We found that helices α4 and α5 of the Get3 TABD are often incompletely resolved, confirming that they are able to adopt different conformations. Notably, one other incidence of state 2 was identified within the crystal structure of *Aspergillus fumigatus* Get3 (*af*Get3; Supplementary Fig. 11A)[9], demonstrating that the Get3 TABD from lower eukaryotes can also adopt this conformation. Interestingly, *af*Get3 crystallised as a trimer of dimers, where α4 of one *af*Get3 dimer interacts with the adjacent dimer (*af*Get3') via α7 and α9 of the TABD[9]. This interaction is a striking mimic of the *hs*Get2 α3' interaction with the *hs*Get3 TABD, which is also mediated by small hydrophobic residues (Supplementary Fig. 11B). In addition, *af*Get3' helix α8 protrudes into the active site of *af*Get3 in a similar manner to the tip of the *hs*Get1-CD (Supplementary Fig. 11C), with both interactions leading to a reconfiguration of the nucleotide binding loops[9,18,19]. Therefore, the overall association of the two *af*Get3 dimers closely resembles that of *hs*Get3 and the cytosolic regions of *hs*Get1/Get2, asserting that these interactions are important for converting the TABD to state 2. Given that the Get3 homodimer is always in state 1 when in complex with just the Get1-CD (Supplementary Data 1), helix α3' is likely to make a more significant contribution than Get1 to this conformational change. Indeed, within both the structures of the *hs*Get3/Get1-CD heterodimer and *ct*GET insertase, Get3 helix α7 is tilted inwards at least as far as the position of *hs*Get2 helix α3' in the wild type human GET insertase structure (Fig. 4C). This suggests that binding of helix α3' pushes this helix within the TABD outwards, which could contribute to the further rearrangements of α4 and α5. Overall, binding of helix α3' to Get3 can be directly correlated with conformational changes in both the TABD and the Get1/Get2 membrane heterotetramer.

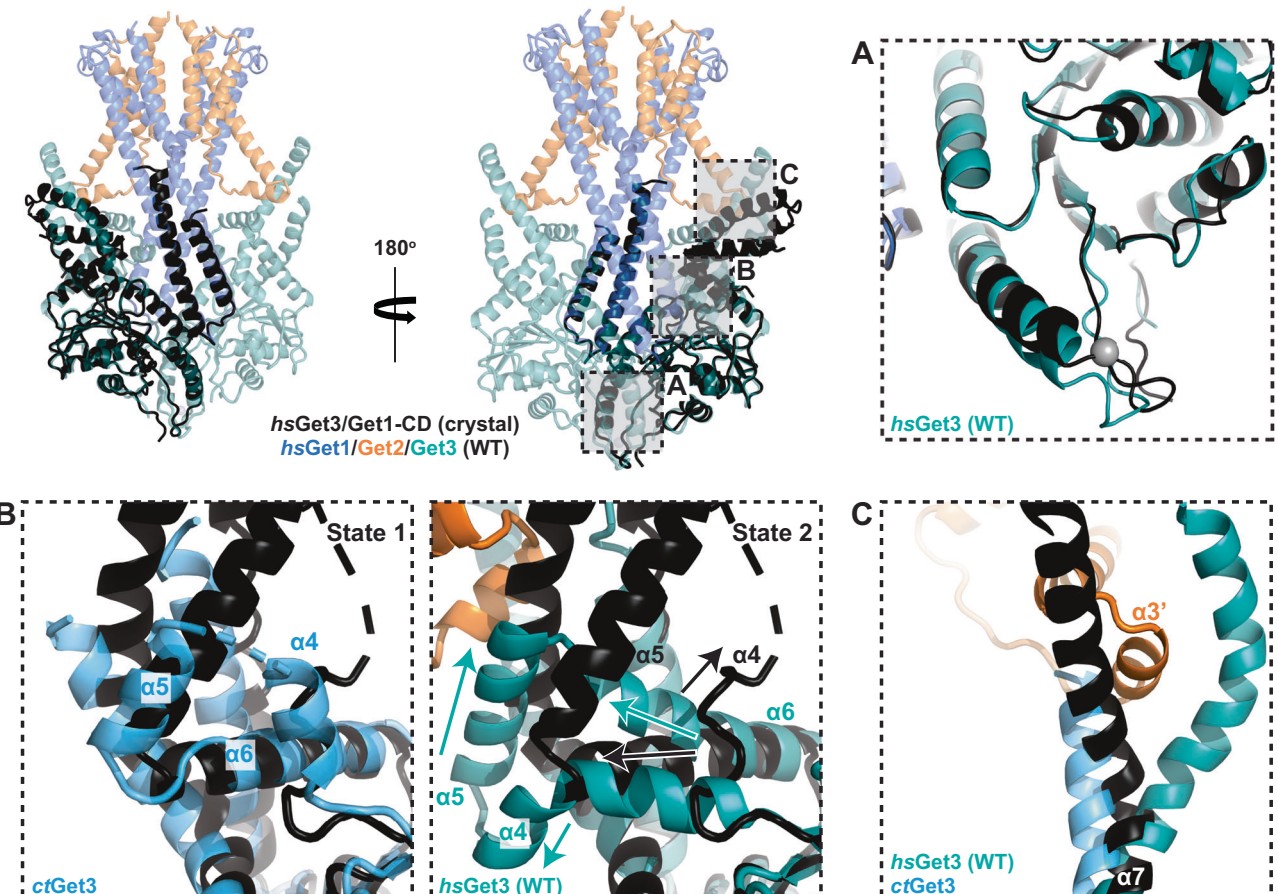

**Fig. 4 | Binding of Get2 helix α3' leads to rearrangement of the Get3 TABD.**
**A–C** Superposition of Get3 within the *hs*Get3/Get1-CD crystal structure (black) with Get3 chain A within the wild type (WT) *hs*Get2^ΔN-Get1/Get3 (RMSD 1.48 Å over 237 Cα atoms) and *ct*Get2^ΔN-Get1/Get3 (RMSD 1.48 Å over 237 Cα atoms) cryo-EM structures. **A** View of the *hs*Get3 dimer interface. The interfacial zinc ion in the WT cryo-EM structure is shown by a grey sphere. **B** View of helices α4-α6 of the Get3 TABD. The arrows show the directionality of the helices. **C** View of helix α7 of the Get3 TABD. Helix α3' is shown from the WT *hs*Get2^ΔN-Get1/Get3 cryo-EM structure.

## The GET insertase induces membrane thinning

A further unexplored aspect of the GET insertase is how the TMDs of Get1/Get2 interact with and influence the structure of the lipid bilayer. Interestingly, the conserved fold of the *ct*Get1/Get2 heterodimer adopts a different position in the membrane compared to the structure of *hs*Get1/Get2, with each heterodimer half exhibiting a 40° rotation with respect to the membrane plane (Fig. 2C and Supplementary Fig. 12A). This is largely due to *ct*Get1 TMD1/TMD2 not adopting the parallel arrangement observed for *hs*Get1, instead being tilted by 40° relative to each other (Supplementary Fig. 12A). Given that the continuous helix formed by TMD2 and the CD is invariant between *ct*Get1 and *hs*Get1, this offset is caused by changes in the tilt of TMD1. This reorientation is likely enabled by the *ct*Get1 AH, which lies almost perpendicular to the membrane and is connected to the CD by three 90° proline/glycine kinks that could serve as hinge regions (Supplementary Fig. 5D). In the *hs*Get1/Get2^Δα3' reconstruction, the asymmetry of the resolved regions is notable given the two-fold symmetry displayed by the wild type complex[24]. Whilst in the resolved *hs*Get1/Get2^Δα3' heterodimer *hs*Get1 TMD1 and TMD2 are still arranged in parallel, in the opposing heterodimer only *hs*Get1 TMD2 is fully resolved, with ambiguous density for *hs*Get1 TMD1 beyond the AH[24] indicating a non-parallel arrangement (Supplementary Fig. 12B). Therefore, TMD1 and the adjoining non-resolved TMDs likely also exhibit positional variability in the plane of the membrane, suggesting that there is plasticity in the connection between *hs*Get1 TMD1 and the AH and that this could represent a common hinge point for movements within the GET insertase.

As a result of the rearrangement of *ct*Get1/Get2, the hydrophilic groove of *ct*Get1/Get2 is tilted so that it is open towards the cytosolic leaflet of the membrane (Supplementary Fig. 12C), in contrast to the *hs*Get1/Get2 hydrophilic groove opening to the cytosol (Supplementary Fig. 12D). In addition, *ct*Get2 TMD3 no longer completely traverses the ER membrane, which is notable given the high number of charged residues found at the N-terminal portion of this helix and the adjoining cytoplasmic loop. Indeed, we observe significant thinning of the nanodisc in the vicinity of *ct*Get2 TMD3 (Fig. 5A). As this helix encloses one side of the hydrophilic groove, such membrane distortion would likely lower the energetic barrier for substrate insertion.

To investigate how the human GET insertase affects membrane morphology, we performed atomistic molecular dynamics (MD) simulations of our structure of *hs*Get2^ΔN-Get1/Get3 embedded in a lipid bilayer of varying lipid compositions (Supplementary Data 2). Each simulation was run for 3 μs to allow sufficient time for the relaxation of the membrane and the protein, and repeated 3 times for better statistics. All simulations revealed that, independent of lipid composition, the membrane was notably thinner in the immediate vicinity of *hs*Get1/Get2 (Fig. 5B and Supplementary Figs. 13 and 14). Interestingly, the cytosolic leaflet was more distorted, particularly at the entrance to the hydrophilic groove (Fig. 5C and Supplementary Movie 1). Specifically, during the course of the MD simulations, we observed that the phospholipid head groups of several lipids are drawn into the hydrophilic groove, leaving their hydrocarbon tails protruding from the groove at an angle of ~60° relative to the membrane normal (Fig. 5C, inset). Interestingly, the *hs*Get1/Get2^Δα3' reconstruction contains an

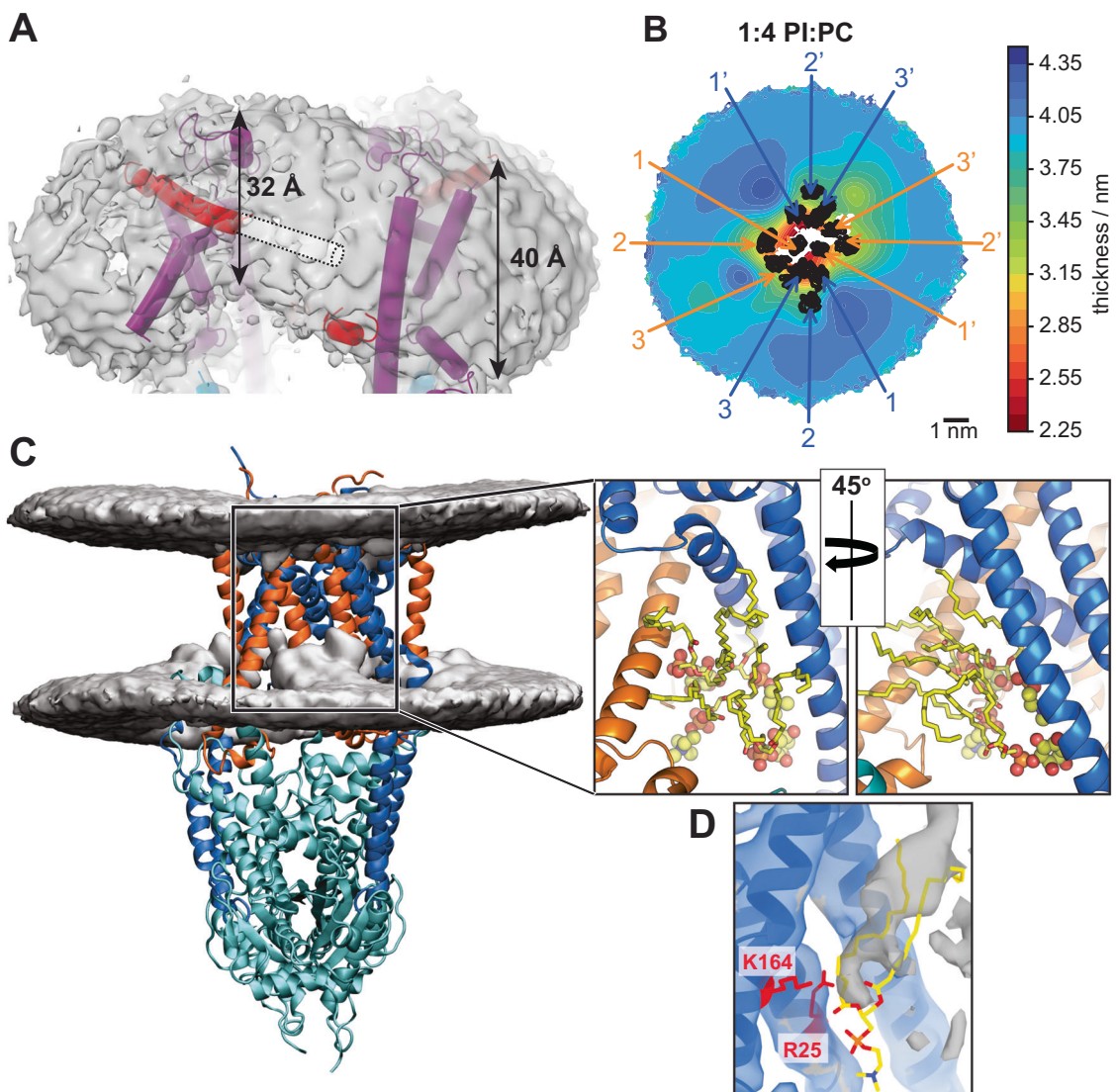

**Fig. 5 | The GET insertase distorts the lipid bilayer. A** Model for nanodisc-embedded *ct*Get2^ΔN^-Get1/Get3 superimposed with the cryo-EM density contoured to 0.092. The dashed cylinder shows the trajectory of the unresolved N-terminal portion of *ct*Get2 TMD3. The approximate thickness of the nanodisc membrane at different positions is indicated. **B, C** Membrane behaviour as captured in atomistic MD simulations of *hs*Get1/Get2 embedded in a 20/80 (mol%/mol%) 16:0-18:1 phosphatidylinositol (PI)/16:0-18:1 phosphatidylcholine (PC) bilayer. **B** Average 2D membrane thickness map constructed based on the local lipid phosphorus atom distance between upper and lower leaflets. The *hs*Get1/Get2 TMDs are labelled within the 2D projection. **C** Average 3D iso-occupancy map for the lipid phosphorus atoms showing local membrane thickness and deformation relative to the structure of *hs*Get1/Get2. Both 2D membrane thickness maps and the 3D iso-occupancy maps were constructed as an average over all 3 simulation repeats after discarding the first 200 ns from each repeat, after the transmembrane region of the protein is aligned on the membrane plane. The zoomed insets show different views of a representative snapshot of the hydrophilic groove at the end of the third trajectory, with a selection of lipids within the groove displayed in yellow (stick representation for tails, sphere representation for head groups). **D** Model for *hs*Get1 superimposed with the *hs*Get2^ΔN/Δα3'^-Get1/Get3 cryo-EM density contoured to 0.17. Examples of hydrophilic residues pointing into the groove are shown in red and the unidentified density is in grey. The hydrophilic groove is overlaid with a phospholipid (yellow) from the representative snapshot of the MD simulation shown in (**B, C**).

unassigned rod of density in the central membrane cavity, which points into the hydrophilic groove of the resolved heterodimer then extends diagonally towards the luminal side of the membrane (Fig. 5D). Our model for the membrane heterotetramer indicates that this density is unlikely to correspond to one of the unassigned TMDs and may rather belong to a factor that has co-purified with the insertase. This factor seems to be amphipathic, given that it is simultaneously embedded in the membrane and also in the direct vicinity of polar residues (e.g. R25 and K164) within the hydrophilic groove. Indeed, this additional density correlates strikingly well with the arrangement of phospholipid species observed in our MD simulations (Fig. 5D), suggesting lipids are most likely to have been trapped in this structure. This lipid arrangement likely encourages the TA protein C-terminal

extension (CTE) to cross the bilayer by providing both a continuous hydrophilic pathway into the hydrophilic groove, and by reducing the hydrophobic distance that must be subsequently traversed. Thus, we conclude that a conserved trait of the GET insertase is to induce membrane thinning, with the hydrophilic groove being sufficient to distort the bilayer. Additionally, our structure indicates that (re)positioning of Get2 TMD3 further disrupts the membrane at the site of TA protein insertion.

## Discussion
Our data show that the fold of the Get1/Get2 heterodimer is conserved from lower to higher eukaryotes and maintains key features such as the hydrophilic groove, helix α3', the ER cap and the AH, indicating that

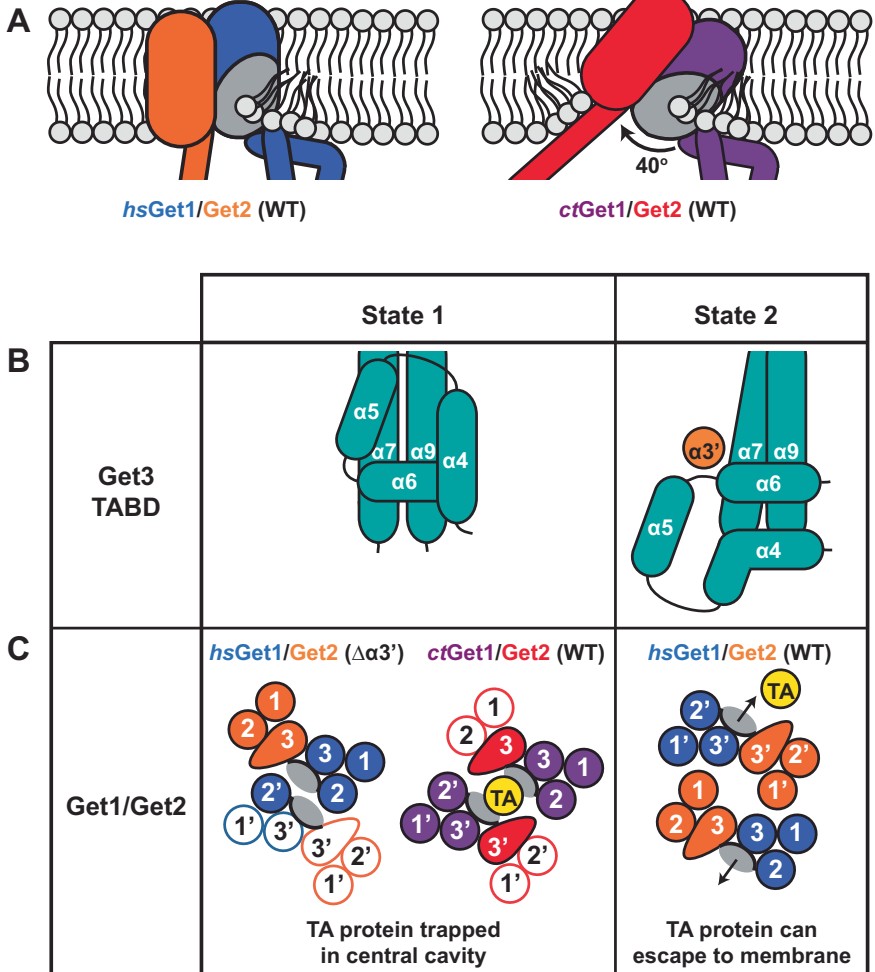

**Fig. 6 | Mechanistic insights into TA protein insertion by the GET insertase.**
**A** The GET insertase mediates membrane thinning through binding of phospholipid head groups to the hydrophilic groove (wild type (WT) hsGet2$^{ΔN}$-Get1/Get3 structure) and repositioning of the Get1/Get2 heterodimer in the membrane plane (WT ctGet2$^{ΔN}$-Get1/Get3 structure). The hydrophilic groove is represented by a grey oval. **B**, **C** The GET insertase exhibits two distinct conformations (state 1 and state 2) dictated by helix α3' interactions. **B** Arrangement of α-helices (teal cylinders) within the Get3 TABD in the absence (state 1) and presence (state 2) of helix α3' binding. **C** Two distinct arrangements of Get1/Get2 TMDs in our three cryo-EM structures as viewed from the ER lumen. TMDs resolved in our structures are shown in solid colour, whilst TMDs modelled based on superimpositions are shown as outlines only. Each hydrophilic groove is represented by a grey oval. In state 1, the TA protein TMD (yellow) would be trapped in a central cavity, whilst in state 2 the TA protein TMD could laterally exit the hydrophilic groove in the direction of the arrows.

the overall mechanism of TA protein insertion is preserved. To this end, we found that the lipid bilayer is thinned in the vicinity of the GET insertase by two structural features (Fig. 6A). Firstly, the hydrophilic groove perturbs the cytosolic leaflet of the membrane by interacting with phospholipid headgroups. Secondly, a putative hinging movement between Get1 TMD1 and the AH repositions the ctGet1/Get2 heterodimer in the membrane plane, drawing ctGet2 TMD3 into the bilayer to further thin the membrane. Thus, the GET insertase effectively lowers the energetic barrier for TA protein insertion by reducing the hydrophobic distance the polar CTE has to cross. Indeed, membrane thinning is a common strategy employed by membrane insertases[30], including the YidC[31,32] and Emc3[33] members of the Oxa1 superfamily.

Our structures also reveal that the GET insertase adopts different conformations (state 1 and state 2), largely dictated by the interactions of Get2 helix α3' within the complex. Binding of helix α3' to Get3 leads to a significant rearrangement of helices α4-α9 within the TABD that form the binding site for the TA protein substrate[13,24] (state 2; Fig. 6B). Therefore, this conformational change likely contributes to TA protein

release to the membrane. Interestingly, Get3 helix α4 was recently found to form an extended loop within the metazoan pre-targeting GET complex[34], whilst helices α4-α6 of *Giardia intestinalis* Get3 undergo extensive rearrangement during client loading and ATP hydrolysis[12]. This provides further evidence that the conformation of this region changes in response to distinct Get3 interaction partners, nucleotide load and the TA protein occupancy.

Although Get1/Get2 homologues from diverse species consistently form a membrane heterotetramer when bound to Get3[24], we observe that this tetramer can adopt two strikingly different morphologies depending on the helix α3' interactions: one where the hydrophilic grooves point inwards towards each other and helix α3' interacts with Get1 TMD2 of the opposing heterodimer (state 1; Fig. 6C) and another where the grooves point outwards to the surrounding membrane and helix α3' interacts with the Get3 TABD (state 2). Within state 2, the Get1-CD is re-oriented relative to all structures of the isolated CD in complex with Get3[18,19,24,35], suggesting it occupies a strained position. We now show that this conformation is governed by binding of helix α3' to the Get3 TABD, and is likely

further stabilised by phosphatidylinositol binding at the heterotetramer interface[24]. Given that a TA protein substrate inserting via the hydrophilic groove would remain trapped in the central membrane cavity of state 1 (Fig. 6C), the formation of state 2 by the GET insertase is presumably required to allow the lateral release of the substrate to the membrane, rationalising the functional importance of helix α3' for TA protein insertion.

It remains to be seen whether state 1 is a bona fide state of the GET insertase or one that is merely observed under non-native conditions. It is easier to envisage that the open conformation of the state 1 heterotetramer in the *ct*Get1/Get2 structure is more likely to lead to the formation of a 2.5 nm channel recently measured for the yeast complex[36], as opposed to the intimate heterotetramer in the human GET insertase structure. However, it is intriguing that helix α3' forms distinct interactions within the wild type *Chaetomium* complex, where loss of lipids during protein purification may have impeded the in vitro stabilisation of state 2. If both conformations of the insertase exist bound to open Get3, it is clear their interconversion cannot occur via a simple linear interpolation, as this leads to severe steric clashes (Supplementary Movie 2). Either this conformational change requires asymmetric movement of the Get1/Get2 heterodimers or it occurs concomitantly with other rearrangements in the insertase, such as hinging between the AH and Get1 TMD1 or opening of the Get3 dimer. Therefore, whilst it is evident that helix α3' is responsible for dynamic changes in the GET insertase, future research will show if, how and when these distinct conformations contribute to TA protein insertion.

## Methods
### Construct design and growth of strains
For protein overexpression, yeast codon-optimised *ct*Get1 and *ct*Get2 sequences were cloned downstream of the GAL1 promoter in the pMT929 vector and expressed in a *S. cerevisiae* Δ*get3* strain[14], all as previously described[24]. Within the *ct*Get2$^{\Delta N}$-Get1 construct, residues 185-357 of *ct*Get2 are fused to *ct*Get1 by a (GS)$_2$-TEV-(GS)$_2$ linker. The *hs*Get2$^{\Delta N/\Delta\alpha3'}$-Get1 construct was prepared from pFastBac1 *hs*Get2$^{\Delta N}$-Get1[24] using the Quikchange Lightning site-directed mutagenesis kit (Stratagene) to replace residues 242-250 of *hs*Get2 with a G$_4$ linker, then expressed in Sf9 insect cells (Thermo Fisher, cat. no. 12659017) as previously described[24]. For expression of *hs*Get1-CD, residues 38-94 of *hs*Get1 were cloned between the NcoI/XhoI sites of the pET24d vector (Novagen) in frame with the C-terminal His$_6$ tag. Residues 14-339 of *ct*Get3 were cloned with a C-terminal Strep-tag II between these sites of pET24d, whilst *hs*Get3 was expressed from the pET24d His$_6$-ZZ-*hs*Get3 construct[24]. *hs*Get1-CD and Get3 sequences were expressed in *E. coli* Rosetta2 (DE3)(Novagen) by autoinduction[37] overnight at 18 °C. pMSP1E3D1 was a gift from Stephen Sligar (Addgene plasmid #20066)[38]. Msp1E3D1 was expressed in *E. coli* BL21 (DE3)(Novagen) grown in Terrific Broth by inducing with 1 mM IPTG for 3 h at 37 °C.

For *S. cerevisiae* growth and functional assays, yeast codon-optimised *ct*Get1 and *ct*Get2 sequences were cloned into the p415 and p416 MET25 plasmids[39] respectively, whilst wild type *hs*Get2 was cloned into p416. p415 *sc*Get1-4PC, p415 *hs*Get1 and p416 *sc*Get2-4PC constructs were used as previously[24]. Site-directed mutagenesis was subsequently used to create mutations in the *get2* sequences. In p416 *ct*Get2$^{\Delta\alpha3'}$ residues 303-313 are replaced by a G$_6$ linker, whilst in p416 *hs*Get2$^{\Delta\alpha3'}$ residues 242-250 are replaced by a G$_4$ linker. p415 *sc*Get1-4PC/p416 *sc*Get2-4PC, p415 *ct*Get1/p416 *ct*Get2 and p415 *hs*Get1/p416 *hs*Get2 variants were co-transformed into a Δ*get1/get2* strain ± the pRS413 GFP-Sed5 plasmid[40] or a Δ*get1/get2 GET3*-GFP strain[14] and grown as previously[24].

### Protein purification
For the purification of Get3 homologues and *hs*Get1-CD, *E. coli* cell pellets were lysed using a M-100L Microfluidizer (Microfluidics) in lysis buffer (50 mM HEPES (pH 7.5), 500 mM NaCl) and the clarified lysate bound to either 2 × 5 ml StrepTrap HP columns (Cytiva; *ct*Get3-Strep) or a 5 ml HisTrap HP column (Cytiva; His$_6$-ZZ-*hs*Get3-Strep and *hs*Get1-CD-His$_6$). The StrepTrap was washed with 20 column volumes (CV) lysis buffer and eluted in 1 CV lysis buffer supplemented with 2.5 mM desthiobiotin, whilst the HisTrap was washed with 20 CV lysis buffer supplemented with 50 mM imidazole. *hs*Get1-CD was eluted with 1 CV lysis buffer supplemented with 250 mM imidazole. *hs*Get3 was eluted by on-column cleavage of the His$_6$-ZZ tag with TEV protease. Affinity-purified constructs were then subjected to size exclusion chromatography (SEC) with a HiLoad 16/60 Superdex 200 pg (Cytiva) column equilibrated in 20 mM HEPES (pH 7.5), 150 mM NaCl (*ct*Get3) or 20 mM HEPES (pH 7.5), 200 mM NaCl, 1 mM TCEP (*hs*Get1-CD and *hs*Get3).

His$_6$-Msp1E3D1 cell pellets were initially lysed in 50 mM HEPES (pH 7.5), 300 mM NaCl, 1% (v/v) Tween-20 and also applied to a 5 ml HisTrap HP column, before washing with 40 CV of the same buffer supplemented with 50 mM sodium cholate, 30 mM imidazole and 0.1% (v/v) Tween-20. His$_6$-Msp1E3D1 was eluted with buffer containing 300 mM imidazole and, without concentrating, subjected to SEC as for the Get3 constructs. In general, protein-containing fractions for His$_6$-Msp1E3D1, Get3 homologues and *hs*Get1-CD were concentrated using a Amicon-Ultra centrifugal filter (Millipore) for long-term storage at −80 °C.

The purification of Get2$^{\Delta N}$-Get1/Get3 complexes for cryo-EM has been described in detail previously[24]. Briefly, His$_8$-tagged *ct*Get2$^{\Delta N}$-Get1 or His$_{10}$-tagged *hs*Get2$^{\Delta N/\Delta\alpha3'}$-Get1 was extracted from the total membranes of *S. cerevisiae* or Sf9 cells respectively in lysis buffer supplemented with 0.5% (w/v) LMNG, then incubated with TALON metal affinity resin (Clontech) for 30 min at 4 °C. This resin was washed with 30 CV lysis buffer supplemented with 20 mM imidazole and 0.01% (w/v) LMNG (Co$^{2+}$ wash buffer), and incubated with 5 mg of the corresponding purified Strep-tagged Get3 homologue in 5 CV Co$^{2+}$ wash buffer for 1 h at 4 °C. The resin was washed again with 15 CV Co$^{2+}$ wash buffer and the protein complex eluted in 5 CV of Co$^{2+}$ wash buffer via on-column cleavage of the tag with His$_6$-tagged HRV-3C Protease. The flow-through was applied to a 1 ml StrepTrap HP column (Cytiva), then washed with 30 CV lysis buffer supplemented with 0.01% (w/v) LMNG, prior to elution with 10 CV of this buffer supplemented with 2.5 mM desthiobiotin. For amphipol exchange, the *ct*Get2$^{\Delta N}$-Get1/Get3 or *hs*Get2$^{\Delta N/\Delta\alpha3'}$-Get1/Get3 eluates were incubated with A835 (Jena Bioscience) or PMAL-C8 (Anatrace) amphipol respectively at a 3x weight excess over the membrane protein fusion for 2 h at 4 °C, followed by 1% (w/v) α-cyclodextrin (Sigma Aldrich) overnight at 4 °C. SEC was then performed with a Superdex 200 10/300 increase 10/30 GL column (Cytiva) equilibrated in 20 mM HEPES (pH 7.5), 200 mM NaCl.

For the reconstitution of *ct*Get2$^{\Delta N}$-Get1/Get3 nanodiscs, 5 mg yeast polar lipid extract was doped with 0.02 mg 16:0 lissamine rhodamine PE (Avanti Polar Lipids) and resuspended to ~30 mM final concentration in 50 mM HEPES (pH 7.5), 500 mM NaCl, 1% (w/v) LMNG. 30 μM *ct*Get2$^{\Delta N}$-Get1/Get3 was mixed with a molar excess of His$_6$-Msp1E3D1 (2x) and lipids (160x) and incubated for 1 h at 4 °C, before supplementing with 0.5% (w/v) α-cyclodextrin (Sigma Aldrich) overnight at 4 °C. SEC was then performed with a Superdex 200 10/300 increase 10/30 GL column (Cytiva) equilibrated in 20 mM HEPES pH7.5, 150 mM NaCl, following the absorbance of protein at 280 nm and lissamine rhodamine at 560 nm.

### Cryo-EM grid preparation and data collection
Purified *ct*Get2$^{\Delta N}$-Get1/Get3 in A835 (2.4 mg/ml), *ct*Get2$^{\Delta N}$-Get1/Get3 in nanodiscs (2.8 mg/ml) or *hs*Get2$^{\Delta N/\Delta\alpha3'}$-Get1 complexes PMAL-C8 (1.2 mg/ml) were concentrated with a VivaSpin Protein Concentrator MWCO 100000 (Cytiva). 3 μl sample was applied to a glow-discharged holey carbon-coated grid (Quantifoil 300 mesh, Cu R2/1), adsorbed for 20 s, blotted for 5 s at 95% humidity and 6 °C before plunge-freezing in

liquid ethane using a Vitrobot Mark IV (Thermo Fisher). All data were collected in counting mode on a 300 kV Titan Krios (Thermo Fisher) on a K3 detector (Gatan). For ctGet2[ΔN]-Get1/Get3 in A835, 5599 movies were collected at 1.375 Å/pixel and 2.45 e⁻/pixel/frame over 40 frames and 4 s total exposure. For ctGet2[ΔN]-Get1/Get3 in nanodiscs, 11,997 movies were collected at 1.11 Å/pixel and 1.56 e⁻/pixel/frame over 48 frames and 2.6 s total exposure. 12,311 images were collected for the hsGet2[ΔN/Δα3']-Get1/Get3 dataset at 1.11 Å/pixel and 0.90 e⁻/pixel/frame over 48 frames and 2.58 s total exposure.

## Image processing

ctGet2[ΔN]-Get1/Get3 amphipol images were processed exclusively using Relion3.0[41] (Supplementary Fig. 4A). Motion correction was carried out with dose-weighting using the Relion3.0 implementation of MotionCorr2[42] and CTF estimation was performed via GCTF[43]. A template for autopicking was generated from 2D classification of 1000 manually picked particles. 4,337,300 ctGet2[ΔN]-Get1/Get3 particles extracted with a 220 × 220 Å box size were used for reference-free 2D classification to select 2,075,169 good particles. An ab initio model calculated using a small subset of particles was used as a template for 3D classification. The best class (796,684 particles) was subjected to several rounds of 3D auto-refinement followed by postprocessing masking and B factor sharpening, resulting in a 5.0 Å final reconstruction.

ctGet2[ΔN]-Get1/Get3 nanodisc (Supplementary Fig. 4B) and hsGet2[ΔN/Δα3']-Get1/Get3 (Supplementary Fig. 2) images were processed using patch motion correction and patch CTF estimation in cryoSPARC v3.2[27]. For ctGet2[ΔN]-Get1/Get3 nanodiscs, 5,218,800 particles were selected using WARP[44], then filtered to 1,139,041 good particles by 2D classification in cryoSPARC. These provided the input for a 3-class ab initio reconstruction and the best class was subjected to eight rounds of heterogeneous refinement against varying junk classes. The 4.7 Å final reconstruction was obtained after a non-uniform refinement with C2 symmetry applied. For hsGet2[ΔN/Δα3']-Get1/Get3, 2,044,854 particles were picked by WARP and 682,175 particles further selected by three rounds of 2D classification. After a 3-class ab initio reconstruction, the best reconstruction was subjected to non-uniform refinement and heterogeneous refinement with no symmetry applied. The 4.2 Å final reconstruction was obtained after a subsequent round of non-uniform refinement.

hsGet2[ΔN]-Get1/Get3 images acquired previously[24] were also re-processed using cryoSPARC v3.2[27] (Supplementary Fig. 6). 1,561,837 particles were re-picked by WARP and filtered to 637,871 particles after two rounds of 2D classification. The best reconstruction from a 3-class ab initio reconstruction was subjected to three cycles of heterogeneous refinement and non-uniform refinement, both with C2 symmetry applied, to achieve the 3.2 Å final reconstruction from 189,844 particles. Both the final ctGet2[ΔN]-Get1/Get3 nanodisc and hsGet2[ΔN]-Get1/Get3 maps were low-pass filtered to 6.5 Å and 6.0 Å respectively to yield more continuous density for flexible, low resolution regions. For all reconstructions, local resolution was estimated using Relion's local postprocessing implementation.

## Structural modelling, refinement and analysis

Model building for ctGet2[ΔN]-Get1/Get3 was initially performed using the amphipol reconstruction. To make an initial model for open ctGet3, each half of the closed ctGet3 dimer (PDB accession 3IQW) was superimposed on open scGet3 (PDB accession 3SJA, RMSD 1.37/1.40 Å over 235/236 Cα atoms for chain A/B respectively). The ctGet3 dimer and scGet1-CD were jiggle-fit into the ctGet2[ΔN]-Get1/Get3 amphipol reconstruction, manually mutated to the homologous ctGet1-CD sequence and repositioned within the density in Coot[45,46]. ctGet1 TMD1/TMD2 were built by helical extension of the CD, with kinks in the region of the amphipathic helix corresponding to the positions of proline/glycine residues. ctGet1 TMD3 and ctGet2 TMD3 were modelled by placing ideal helices in the density and assigning the sequence based on sequence alignment with PRALINE[47], TMD prediction with Phobius[48] and secondary structure prediction with PSIPRED[49], with the clear density for the aromatic stacking between W161 and F347 fixing the helical register. Subsequently, an AlphaFold[29] model for the ctGet1/Get2 heterodimer was generated using the ColabFold Alpha-Fold2_advanced Jupyter notebook inside Google Colaboratory[50]. This model confirmed the register of our experimentally derived model (Supplementary Fig. 7) and allowed the ER cap to be modelled in the density. Our complete model for ctGet2[ΔN]-Get1/Get3 in amphipol was then jiggle-fit into the final ctGet2[ΔN]-Get1/Get3 nanodisc reconstruction in Coot.

Within the improved hsGet2[ΔN]-Get1/Get3 reconstruction, improved density for bulky side chains within hsGet2 TMD3 and disulphide bonds to hsGet2 TMD1 and hsGet1 TMD1 confirmed the register of the published model (PDB accession 6SO5)[24]. Whilst minor differences in the register of the ColabFold model generated for the hsGet1/Get2 heterodimer did not fully satisfy these density features (Supplementary Fig. 7A), it was used to derive a model for the ER cap to update the hsGet2[ΔN]-Get1/Get3 model. In order to build the model for hsGet2[ΔN/Δα3']-Get1/Get3, the structure of hsGet2[ΔN]-Get1/Get3 (PDB accession 6SO5) was docked into the density for hsGet3 using Coot. hsGet1/Get2 heterodimers were initially jiggle-fit into the density as rigid bodies and parts of the structure unresolved in the density map were removed from the model. Individual helices were then manually repositioned within the density in Coot.

For all cryo-EM structures, real-space refinement in Phenix[51] with secondary structure and geometry restraints produced the final models (Table 1). Although membrane protein fusions were used, the amino acid residues are numbered relative to the start of the native ctGet1/Get2 and hsGet1/Get2 sequences. Structural figures were prepared in UCSF ChimeraX[52] and PyMOL (Schrödinger, LLC). Superimpositions were performed with Superpose[53]. Videos were prepared in UCSF ChimeraX[52].

## SEC-MALS

SEC-MALS was performed using an ÄKTA™ purifier (Cytiva) coupled to a DAWN® Heleos II 8 + MALS detector and an Optilab® T-rEX dRI monitor (Wyatt Technology). A Superdex 200 10/300 increase 10/30 GL column (Cytiva) was equilibrated with at least 4 CV of 20 mM HEPES, pH 7.5, 200 mM NaCl and 1 mM TCEP. 80 μM hsGet3 was incubated with 800 μM hsGet1-CD for 1 h at room temperature, before 100 μL was injected. Data analysis was performed using Astra 6 (Wyatt Technology) assuming a dn/dc value of 0.185 ml/g.

## Crystallisation and structure determination of the hsGet3/Get1-CD complex

The hsGet3/Get1-CD complex was crystallised at 18 °C by the vapour diffusion sitting-drop method. To obtain well-diffracting crystals, a mixture of 15 mg/ml hsGet3 and 4 mg/ml hsGet1-CD was present in a 450 nl drop in a 1:2 volumetric ratio with 44% (v/v) 1,2-propanediol, 0.05 M calcium acetate and 0.1 M sodium acetate (pH 5.0). Crystals were cryo-protected with 1:4 (v/v) ethylene glycol:mother liquor and flash frozen in liquid nitrogen. Diffraction images were acquired from a single crystal at beamline ID23-1 of the European Synchrotron Radiation Facility (ESRF, Grenoble, France)(7200 images, 0.05° oscillation range, 0.05 s exposure time, 360 s total exposure, 12.750 keV). Data were integrated using XDS[54] and scaled using AIMLESS[55] and STARANISO within the autoPROC toolbox[56]. A cut-off resolution of 2.8 Å was determined for the crystal belonging to the space group P2₁2₁2₁. Phasing by molecular replacement was performed using the hsGet2[ΔN]-Get1/Get3 model (PDB accession 6SO5). The model was rebuilt and refined iteratively using Coot[46] and Phenix[51], until a final model was

reached with $R_{work}/R_{free}$ of 0.24/0.28 and 96.7%/3.3% residues in the favoured/allowed regions of the Ramachandran plot (Table 2). Structural figures were prepared in PyMOL (Schrödinger, LLC).

## Yeast cell lysis for protein analysis

Sample preparation was adapted from the previously described NaOH lysis protocol[57]. Briefly, 750 μl of logarithmically growing cells were pelleted and then resuspended in 1 ml 250 mM NaOH. Samples were incubated on ice for 10 min, pelleted for 1 min by centrifugation at 16,000 g, and resuspended in NuPAGE LDS Sample buffer (Thermo Fisher Scientific) corresponding in μl to 100 × OD600 of the NaOH solution containing the samples. After incubation at 70 °C for 5 min, the samples were centrifuged at 16,000 g for 30 s and stored at −20 °C until later use. A total of 7 μl was used for Western blot analysis.

## Western blotting

Samples were resolved in Bis-Tris gels and transferred onto PVDF membranes. The membrane was blocked in Tris-buffered saline (TBS) containing 5% milk for 1 h followed by incubation with primary antibodies (anti-*hs*Get1 rabbit polyclonal at 1:500 (Synaptic Systems), anti-*hs*Get2 guinea pig polyclonal at 1:2000 (Synaptic Systems) or anti-Pgk1 mouse monoclonal at 1:10000 (Thermo Fisher Scientific)) in TBS containing 0.1% Tween-20 (Roth) overnight at 4 °C. After rinsing the membranes in TBS containing 0.1% Tween-20, incubation with the secondary antibodies followed in TBS containing 0.1% Tween-20 and 0.01% sodium dodecyl sulfate. Membranes were scanned in a LI-COR Odyssey scanner.

## Functional assays in S. cerevisiae

For the analysis of *sc*Get1-4PC/Get2-4PC, *ct*Get1/Get2 and *hs*Get2$^{\Delta\alpha3'}$ variants expressed in *S. cerevisiae* strains, plate growth assays, microscopy of fluorescent strains and the quantification of GFP-Sed5 images were carried out as previously described[24].

## Sequence alignment of Get1 homologues

Get1 homologues were identified using a blast search among fungi based on the amino acid sequence of *sc*Get1 and among animals based on *hs*Get1, yielding 453 homologues among fungi and 445 among animals (vertebrates and arthropods). Sequences were aligned using Clustal Omega[58], then visualized and manually adjusted in Jalview[59]. Logos of consensus sequences were generated using WebLogo 3[60].

## Atomistic molecular dynamics simulations

The initial model for the *hs*Get2$^{\Delta N}$-Get1/Get3 complex used in simulations was constructed based on the cryo-EM structure (PDB accession 6SO5). Missing residues (except the terminal ones) were modelled using Modeller[61]. Since the structure consists of dimers, the structural information from the more complete chain was used in the modelling where possible without enforcing symmetry. Disulphide bridges captured in the cryo-EM structure were also included: two between *hs*Get3 dimers, one between each pair of *hs*Get1 and *hs*Get2, and one in each *hs*Get2. To improve the quality of the modelled segments of the final model, real space refinement was performed using Phenix[51]. The final model was used as the basis for all simulations (Supplementary Data 2).

Using CHARMM-GUI[62], all eight simulation systems (Supplementary Data 2) were constructed in which the lipid composition of the lipid bilayer was varied. To describe the interactions, we used the Charmm36(m) force field for lipids[63] and the protein[64], the TIP3P model[65] for water molecules, and a compatible parameter set for the ions[66]. This atomistic force field parameter set has been designed to investigate protein and lipid interactions[63] and have been successfully used in earlier studies[67]. For each system, the protein was embedded in a membrane. The numbers of lipids and water were adjusted to obtain a sufficiently large hexagonal prism box with a base edge of ~110 Å and

a height of ~170 Å. Each system was solvated with 150 mM KCl and neutralized by additional counter ions. WYF parameters[68] were used for improved cation-π interactions and the Hydrogen Mass Repartitioning (HMR) method[69] was applied to increase the integration time step to 4-fs. Each simulation system was equilibrated following the CHARMM-GUI protocol before the production runs. The duration of a single production simulation for each system was 3 μs, and they were repeated for each system three times. The total simulation time was therefore 72 μs.

All simulations were performed using GROMACS 2020[70]. The equations of motion were integrated using a leap-frog algorithm with a 4-fs time step by the virtue of HMR All bond lengths were constrained using the LINCS algorithm[71]. All three dimensions were treated with periodic boundary conditions. For Coulombic interactions, a real space cut-off of 1.2 nm was used and the long-range electrostatic interactions were computed using the fast smooth Particle-Mesh Ewald (SPME) method[72] with a Fourier spacing of 0.12 nm and a fourth-order interpolation. For the van der Waals interactions, a Lennard-Jones potential with a force-switch between 1.0 and 1.2 nm was used. All production simulations were performed in the NpT ensemble. The Nosé-Hoover thermostat[73,74] was used to maintain the temperature at 310 K, where the protein, the membrane and the solvent (water and KCl) were coupled to separate temperature baths with a time constant of 1.0 ps. The Parrinello-Rahman barostat[75,76] was used for semi-isotropic pressure coupling at 1 atm with a time constant of 5 ps and a compressibility value of $4.5 \times 10^{-5}$ bar$^{-1}$.

The two-dimensional (2D) membrane thickness plots and the three-dimensional (3D) iso-occupancy plots were performed after discarding the first 200 ns of each simulation. Supplementary Fig. 14 shows that during this 200 ns period the thickness of the membrane adapts to the protein confirming sufficient equilibration, and that the results are independent of the starting configuration. To ensure a common reference, all trajectories were post-processed to superpose the transmembrane portion of the protein to that of the crystal structure on the membrane (*xy*) plane. The 2D thickness maps were generated using in-house scripts. First, for each frame, smooth interpolations (Clough-Tocher) of the *z*-coordinates of the phosphorus atoms were calculated on a grid on the *xy*-plane separately for the upper and lower leaflets. Then, the interpolated surfaces were subtracted from each other, and averaged over time and the simulation repeats. The 2D lateral densities of the transmembrane helices were overlayed onto thickness maps to visualize the location of the protein in the membrane. The Visual Molecular Dynamics (VMD) program[77] was used for visualization, snapshots and movies, as well as generation of the occupancy iso-surfaces (iso-occupancy value of about 0.6).

## Reporting summary

Further information on research design is available in the Nature Portfolio Reporting Summary linked to this article.

## Data availability

Coordinates for *hs*Get3/Get1-CD, *hs*Get2$^{\Delta N}$-Get1/Get3, *hs*Get2$^{\Delta N/\Delta\alpha3'}$-Get1/Get3, *ct*Get2$^{\Delta N}$-Get1/Get3 in amphipol and *ct*Get2$^{\Delta N}$-Get1/Get3 in nanodiscs have been deposited in the Protein Data Bank under accession codes 8CQZ (*hs*Get1-CD/Get3 crystal structure), 8CR1 (*hs*Get2$^{\Delta N}$-Get1/Get3), 8CR2 (*hs*Get2$^{\Delta N/\Delta\alpha3'}$-Get1/Get3), 8ODU (*ct*Get2$^{\Delta N}$-Get1/Get3 in amphipol), and 8ODV (*ct*Get2$^{\Delta N}$-Get1/Get3 in nanodiscs) respectively. The respective cryo-EM volumes have been deposited in the Electron Microscopy Data Bank under accession codes EMD-16801, EMD-16802, EMD-16817 and EMD-16819. The source data underlying Supplementary Fig. 9B is provided as a Source Data file. All MD simulation data are uploaded to zenodo.org with the https://doi.org/10.5281/zenodo.8420199. Source data are provided with this paper.

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

## Acknowledgements

We thank Jürgen Kopp and Claudia Siegmann from the BZH/Cluster of Excellence: CellNetworks crystallisation platform and acknowledge access to beamline ID23-1 at the ESRF in Grenoble and the support of the beamline scientists. We acknowledge access to the infrastructure of the Cryo-EM Network at the Heidelberg University (HDcryoNET) and support by Dirk Flemming (BZH) and Götz Hofhaus (Bioquant). We also acknowledge the services SDS@hd and bwHPC supported by the Ministry of Science, Research and the Arts Baden-Württemberg (MWK) and the Deutsche Forschungsgemeinschaft (DFG) through grants INST 35/1314-1 FUGG and INST 35/1134-1 FUGG. We thank Stephen Sligar for the pMSP1E3D1 plasmid. This work was supported by the DFG through the Leibniz Programme (SI 586/6-1) and TRR83 (TP22) to I.S. Work by I.V. and G.E. was supported by the Academy of Finland (331349, 336234, 346135), the Sigrid Juselius Foundation, the Helsinki Institute of Life Science (HiLIFE) Fellow Program, the Human Frontier Science Program (RGP0059/2019) and the DFG through TRR83. A.F. and B.S. were funded by the DFG through SFB1190 (P04, Projektnummer 264061860).

## Author contributions

M.A.M., M.H. and I.S. designed the study, analysed data and interpreted the results. M.A.M. and M.H. performed purification of cryo-EM samples, collected EM data, processed EM data and built structural models. M.H. and D.S. performed purification of samples for crystallisation, MALS, crystallographic structure determination and model building. K.W. performed refinement and analysis of structural models. G.E. and I.V. performed and analysed atomistic molecular dynamics simulations. A.F. and B.S. performed and analysed functional assays in yeast and the sequence alignment of Get1 homologues. M.A.M. and I.S. wrote the manuscript. All authors contributed to the final version of the manuscript.

## Funding

## Competing interests

The authors declare no competing interests.
