## [Peer Review File · Nature Communications]

The GET insertase exhibits conformational plasticity and induces membrane thinningReviewers' Comments:

Reviewer #1:

Remarks to the Author:

The manuscript from Sinning et al. details new structures, simulations, and biochemical data on proteins in the GET pathway, which mediates insertion of tail-anchored membrane proteins in the ER. Comparisons between structures and data from the human and fungal Get1/2/3 reveal important similarities and differences.

Overall, I think there is a lot of interesting data here and I commend the authors for their hard work on this system. However, I feel like the data have not come together into a very coherent story. I found it difficult to follow the different claims made, which are often extrapolated a bit too far from the underlying data.

For example, the structure in the human Get1/2 complex is relatively low resolution (4.2 Å) and incomplete, complicating interpretation. Extra density is suggested to not belong to a missing TM but rather to a co-purified factor, which is then assumed to be amphipathic. This is then taken to confirm that the hydrophilic groove can interact with the polar aspect of such molecules. Even if this is reasonable, it's based on a chain of assumptions and not strongly supported by the data.

I think Table S2 is missing?

Reviewer #2:

Remarks to the Author:

An important unresolved issue in the Get pathway is the mechanism of tail-anchored proteins insertion into the bilayer by the Get1/2 complex. Central to this process is an evolutionarily conserved hydrophilic groove present in the Get1 subunit and other members of the Oxa1 superfamily of insertases. But exactly how the Get1/2 complex coordinates release of a tail-anchored substrate from the Get3 targeting factor and then inserts it into the membrane, remains unclear. In particular, it is not known whether one or two copies of the Get1/2 complex binds to homodimeric Get3. Similarly, it is not known whether the route into the membrane involves one or two copies of the Get1/2 complex. Previous work from the authors described a cryoEM structure of human Get1/2/3 comprising two copies of each protein (a 2:2:2 complex), while single-molecule studies suggest that the minimal functional unit comprises one Get1/2 complex bound to the Get3 homodimer (1:1:2 complex). Distinguishing between these two possibilities is important for understanding the mechanism of tail-anchored protein release and insertion by the Get pathway.

Taking advantage of N-terminally truncated single-chain hsGet2-Get1 construct used previously, the authors examine the effect of replacing the Get2 helix $\alpha 3'$ motif with polyglycine. This helix was shown previously to make important contacts with the substrate binding region of Get3. Compared to the original structure, mutation of helix $\alpha 3'$ dramatically disrupts the tetrameric Get1/2 interface, with density for one of the two Get1/2 complexes largely disordered. Next a structure of *C. thermophilum* Get1/2/3 is determined, once again using a construct that lacks the entire N-terminal region of Get2. As expected, the overall fold of the Get1 and Get2 subunits is similar to those observed previously for the human and yeast homologs. However, the Get2 subunits are largely disordered except for TMD3 which assumes an unusual orientation with respect to the plane of the membrane. Moreover, compared with the original hsGet1/2/3 structure, the *C. thermophilum* Get1/2 tetramer interface is mostly disrupted and dramatically rearranged—whereas in the human Get1/2/3 complex the conserved hydrophilic grooves of each Get1 subunit face “outward” (state 2), in *C. thermophilum*, the grooves face each other, forming a central membrane cavity (state 1). This leads the authors to propose a model in which a tail-anchored substrate first enters the membrane via the central cavity (state 1), and is subsequently released to the bilayer following a rearrangement of the Get1/2

tetramer into state 2. As noted by the authors, however, it is difficult to envision how the Get1/2 tetramer could interconvert between these two states because of severe steric clashes between Get1/2 dimers. Finally, the authors carry out MD simulations, finding that the membrane is distorted and thinned by Get1/2 complex. Similar to what has been proposed for other members of the Oxa1 group, such thinning is thought to lower the energetic barrier to translocation of hydrophilic segments across the membrane.

From a technical standpoint, this work is well done. My major concern, however, is that the authors are attempting to rationalize a series of unusual structures in the context of a 2:2:2 complex, without first demonstrating that this is, in fact, the relevant stoichiometry. While it is possible that these different structures represent snapshots of a highly dynamic complex that undergoes wholesale subunit rearrangements, it is also possible that they are an artifact of the constructs and/or experimental conditions used for structure analysis. In this regard, it is notable that these Get1/2/3 structures were determined using constructs in which the N-terminus of Get2 is deleted. Importantly, this region contains a high-affinity binding site for Get3. In the absence of this, the binding sites on opposite sides of the Get3 homodimer can be occupied by two Get1 subunits. Whether the resulting 2:2:2 arrangement is physiologic, or merely an artifact arising from truncating the Get2 N-terminus and purifying in detergent, remains ambiguous. This is a significant concern given that the oligomeric state of purified Get1/2 varies (dimer, tetramer) depending on multiple factors, including: type of detergent used for purification, mutations that disrupt one or more of the high affinity binding sites, the presence or absence of Get3, etc. as shown previously by the authors (McDowell et al. Mol Cell 2020). While the author's state 1/state 2 model could in principle be a reasonable proposal for a 2:2:2 complex (notwithstanding concerns about how the two states could interconvert), it is not meaningful for a 1:1:2 complex. Because the data shed no light on this critical question, the manuscript's impact is modest—mainly limited to the observation that *C. therm.* Get1, Get2 and Get3 adopt similar fold to the yeast and human homologs, and molecular dynamics simulations that support membrane thinning near Get1/2.

Other comments:

*Are residues lining the tetrameric interface(s) conserved as would be expected if these surfaces are functionally relevant?

*The hsGet1/Get2/Get3 should not be referred to as "WT", since it is a mutant (~150 N-terminal residues have been removed) that may not represent the physiologically relevant state of the functional, full-length Get1/2/3 complex.

Reviewer #1 (Remarks to the Author):

The manuscript from Sinning et al. details new structures, simulations, and biochemical data on proteins in the GET pathway, which mediates insertion of tail-anchored membrane proteins in the ER. Comparisons between structures and data from the human and fungal Get1/2/3 reveal important similarities and differences.

Overall, I think there is a lot of interesting data here and I commend the authors for their hard work on this system.

We thank the reviewer for these positive comments.

However, I feel like the data have not come together into a very coherent story.

We apologise that the reviewer finds the story incoherent and we have revised the manuscript to link the different themes better. Ultimately our story resolves several questions that arise from the data presented in McDowell et al., 2020 and arrives at three major conclusions:

- The structure of the Get1/Get2 heterodimer is largely conserved across eukaryotes.
- Helix $\alpha 3'$ binding to Get3 drives a distinct conformation of both the Get3 TA binding domain and of the Get1/Get2 membrane heterotetramer, both of which are likely to be important for TA protein release and insertion.
- The GET insertase causes membrane thinning, a common mechanism used by membrane insertases to reduce the energetic barrier for protein insertion.

I found it difficult to follow the different claims made, which are often extrapolated a bit too far from the underlying data.

Again, we regret that the reviewer has this impression. We have now rearranged several points in the manuscript to ensure each conclusion is presented alongside all its supporting data. However, this is also a general criticism, which is difficult to respond to. If the reviewer gives more specific examples, we could better address this comment, as we have done below for the one specific example given.

For example, the structure in the human Get1/2 complex is relatively low resolution (4.2 Å) and incomplete, complicating interpretation.

The resolution and completeness are sufficient to determine that deletion of helix $\alpha 3'$ leads to a complete rearrangement and loss of symmetry in the membrane heterotetramer. These global rearrangements are the take-home message from the structure, and we do not attempt to interpret details that would require a higher resolution. The observed destabilization and rearrangements of the Get1/2/3 complex by deletion of helix $\alpha 3'$ is really intriguing and could not be expected from our previous structure. It also shows that the structural landscape of the GET insertase is by far larger than envisioned, and knowing this allows now to design experiments to understand the regulation of the insertase in a direction that was not possible before.

Extra density is suggested to not belong to a missing TM but rather to a co-purified factor, which is then assumed to be amphipathic. This is then taken to confirm that the hydrophilic groove can interact with the polar aspect of such molecules. Even if this is reasonable, it's based on a chain of assumptions and not strongly supported by the data.

The molecular dynamics simulations provide evidence that hydrophilic head-groups of lipids can occupy the hydrophilic groove in exactly the orientation of our density. We have rearranged the text so that the extra density appears in the same section as the molecular dynamics simulations.

I think Table S2 is missing?

We thank the reviewer for noticing this and indeed realise we unintentionally omitted it during submission. It has been provided with the revised manuscript.

Reviewer #2 (Remarks to the Author):

An important unresolved issue in the Get pathway is the mechanism of tail-anchored proteins insertion into the bilayer by the Get1/2 complex. Central to this process is an evolutionarily conserved hydrophilic groove present

in the Get1 subunit and other members of the Oxa1 superfamily of insertases. But exactly how the Get1/2 complex coordinates release of a tail-anchored substrate from the Get3 targeting factor and then inserts it into the membrane, remains unclear. In particular, it is not known whether one or two copies of the Get1/2 complex binds to homodimeric Get3. Similarly, it is not known whether the route into the membrane involves one or two copies of the Get1/2 complex. Previous work from the authors described a cryoEM structure of human Get1/2/3 comprising two copies of each protein (a 2:2:2 complex), while single-molecule studies suggest that the minimal functional unit comprises one Get1/2 complex bound to the Get3 homodimer (1:1:2 complex). Distinguishing between these two possibilities is important for understanding the mechanism of tail-anchored protein release and insertion by the Get pathway.

We thank this referee for the comments on our manuscript. We agree, the oligomeric state of Get1/Get2 has been a controversial and non-trivial question in the field for years and one which many groups have tried to address using various methods. As outlined above in the comment, our previous work (McDowell et al. 2020) uses a combination of cryo-EM, native mass spectrometry and functional assays to describe the 2:2:2 complex, providing the starting point for this manuscript. The above also mentioned single-molecule studies refers to the publication by Zalisko et al. (2017), however there is also recent important evidence from Heo et al., 2023 (<https://doi.org/10.1016/j.celrep.2022.111921>) not mentioned. This study shows that Get1/2 forms a channel in a membrane environment, comprising two Get1/2 heterodimers (as in our structures) - suggesting a 2:2:2 complex. In addition, the Get1 cytoplasmic domain displaces the Get2 cytoplasmic domain from its binding site on Get3 and is symmetrically bound in structures with Get3, even when Get2 is present (Stefer et al., 2011). Whilst the Zalisko et al., 2017 paper is the only study so far providing evidence for a 1:1:2 complex, our data do not preclude that this complex can act as the minimal functional unit as the site of protein insertion (the hydrophilic groove) is formed by one Get1/Get2 heterodimer. However, evidence from McDowell et al., 2020 shows that the 2:2:2 complex is formed and enhances the efficiency of TA protein insertion.

Taking advantage of N-terminally truncated single-chain hsGet2-Get1 construct used previously, the authors examine the effect of replacing the Get2 helix a3' motif with polyglycine. This helix was shown previously to make important contacts with the substrate binding region of Get3. Compared to the original structure, mutation of helix a3' dramatically disrupts the tetrameric Get1/2 interface, with density for one of the two Get1/2 complexes largely disordered. Next a structure of C. thermophilum Get1/2/3 is determined, once again using a construct that lacks the entire N-terminal region of Get2. As expected, the overall fold of the Get1 and Get2 subunits is similar to those observed previously for the human and yeast homologs. However, the Get2 subunits are largely disordered except for TMD3 which assumes an unusual orientation with respect to the plane of the membrane. Moreover, compared with the original hsGet1/2/3 structure, the C. thermophilum Get1/2 tetramer interface is mostly disrupted and dramatically rearranged—whereas in the human Get1/2/3 complex the conserved hydrophilic grooves of each Get1 subunit face “outward” (state 2), in C. thermophilum, the grooves face each other, forming a central membrane cavity (state 1). This leads the authors to propose a model in which a tail-anchored substrate first enters the membrane via the central cavity (state 1), and is subsequently released to the bilayer following a rearrangement of the Get1/2 tetramer into state 2. As noted by the authors, however, it is difficult to envision how the Get1/2 tetramer could interconvert between these two states because of severe steric clashes between Get1/2 dimers.

In the discussion, we were careful not to propose a model where state 1 converts to state 2, but apologise if this has been misinterpreted. The major message of the paper is that we observed two states of the Get1/2 tetramer, and these are dictated by whether helix a3' interacts with Get3. Thus, we define a further role for this functionally important element in determining the conformation of the Get1/2 tetramer. We further note that state 2 is likely to be functionally important, as the hydrophilic grooves point outwards and would allow release of the tail-anchored protein substrate to the membrane. However, from our discussion, “*it remains to be seen whether state 1 is a bona fide state of the GET insertase or one that is merely observed under non-native conditions.*” We also note that interconversion between the states cannot occur by linear interpolation, however it is not impossible, requiring either asymmetric movement of the Get1/2 heterodimers or concomitant rearrangements within the Get3 dimer or Get1. The identification of state 1 is therefore novel, but its significance is a topic for subsequent research.

Finally, the authors carry out MD simulations, finding that the membrane is distorted and thinned by Get1/2 complex. Similar to what has been proposed for other members of the Oxa1 group, such thinning is thought to lower the energetic barrier to translocation of hydrophilic segments across the membrane.

From a technical standpoint, this work is well done. My major concern, however, is that the authors are attempting to rationalize a series of unusual structures in the context of a 2:2:2 complex, without first demonstrating that this is, in fact, the relevant stoichiometry.

As outlined above, we previously showed the 2:2:2 complex is a relevant stoichiometry in McDowell et al., 2020 and recent evidence from Heo et al., 2023 also indicates this. Indeed, these peer-reviewed works provide the starting point for the work presented in this manuscript. We are not sure what is meant by ‘unusual’, but we think the conformations of our structures are exactly what makes them intriguing. For example, the observed destabilization and rearrangements of the Get1/2/3 complex by deletion of helix a3’ could not be expected from our published structure and implicates a key role for this element in dictating the conformation of the GET insertase.

While it is possible that these different structures represent snapshots of a highly dynamic complex that undergoes wholesale subunit rearrangements, it is also possible that they are an artifact of the constructs and/or experimental conditions used for structure analysis. In this regard, it is notable that these Get1/2/3 structures were determined using constructs in which the N-terminus of Get2 is deleted. Importantly, this region contains a high-affinity binding site for Get3. In the absence of this, the binding sites on opposite sides of the Get3 homodimer can be occupied by two Get1 subunits. Whether the resulting 2:2:2 arrangement is physiologic, or merely an artifact arising from truncating the Get2 N-terminus and purifying in detergent, remains ambiguous.

Evidence in McDowell et al., 2020 already indicates this is not the case: complexes containing full-length yeast Get2 were demonstrated to also tetramerise in the presence of Get3, using both native mass spectrometry and blue native gels of complexes isolated from endogenous membranes. We showed that this complex has a 2:2:4 stoichiometry, with the cytoplasmic domain of Get2 symmetrically binding to an additional Get3 dimer. Furthermore, we have since performed cryo-EM and native MS of full length human Get1/2 in complex with Get3 and find that Get3 is bound to a Get1/2 heterotetramer via symmetric binding of two Get1 cytoplasmic domains, as observed in all our structures where Get2 has a truncated N-terminus.

This is a significant concern given that the oligomeric state of purified Get1/2 varies (dimer, tetramer) depending on multiple factors, including: type of detergent used for purification, mutations that disrupt one or more of the high affinity binding sites, the presence or absence of Get3, etc. as shown previously by the authors (McDowell et al. Mol Cell 2020).

In McDowell et al., 2020, we showed that certain detergents stabilize the tetramer in the absence of Get3 as they allow co-purification of the lipid phosphatidylinositol, which we demonstrated binds to the tetramer interface of Get1/2. However, we showed that Get1/2 forms a tetramer in the presence of Get3, regardless of the detergent used for purification, as is the condition used in this manuscript. Mutation of the Get1 high affinity binding site was used to show that two copies of full-length Get2 within the Get1/2 tetramer are capable of symmetric binding to an independent Get3 dimer.

While the author's state 1/state 2 model could in principle be a reasonable proposal for a 2:2:2 complex (notwithstanding concerns about how the two states could interconvert), it is not meaningful for a 1:1:2 complex. Because the data shed no light on this critical question, the manuscript's impact is modest—mainly limited to the observation that C. therm. Get1, Get2 and Get3 adopt similar fold to the yeast and human homologs, and molecular dynamics simulations that support membrane thinning near Get1/2.

While it is shown in this manuscript, McDowell et al., 2020 and Heo et al., 2023 that the 2:2:2 stoichiometry is conserved from different fungi to humans, the observed destabilization and rearrangements of the Get1/2/3 complex by deletion of helix a3’ are unexpected and intriguing, which shows that the structural landscape of the GET insertase is by far larger than envisioned. We also provide first evidence that the GET insertase follows a “general” mechanism for membrane insertion via membrane thinning as seen for members of the Oxa1 family of insertases. Taken together, we see this as an important starting point in the field for further studies.

Other comments:

**Are residues lining the tetrameric interface(s) conserved as would be expected if these surfaces are functionally relevant?*

In general, sequence conservation within the transmembrane domains of Get1/2 is very low between lower and higher eukaryotes (especially for Get2, where the human homologue was eventually identified by functional similarities). Despite this, the core structure and function are remarkably conserved. Therefore, we are not surprised there is not strict sequence conservation at the tetramer interface comprising Get2. However, we previously showed that positively charged residues are found within the interfacial helices of both *H. sapiens* and *S. cerevisiae* Get2, and that their mutation leads to impairment of phosphatidylinositol binding, tetramerisation and tail-anchored protein insertion (McDowell et al., 2020). Therefore, positively charged

residues within the tetramer interface are functionally relevant in diverse species. In addition, we have shown that further elements also contribute to formation of the intimate membrane heterotetramer (Get3 binding to Get1/Get2, helix a3' binding to Get3).

**The hsGet1/Get2/Get3 should not be referred to as "WT", since it is a mutant (~150 N-terminal residues have been removed) that may not represent the physiologically relevant state of the functional, full-length Get1/2/3 complex.*

We appreciate the reviewer's concern, although we were aiming to cause minimal confusion to the reader when differentiating between the human variants with native and deleted helix a3'. Therefore, we have now clearly defined what we mean by 'wild type' in the context of this study at the beginning of the results section.

Reviewers' Comments:

Reviewer #1:

Remarks to the Author:

The presentation is much improved. The paper merits publication.

Reviewer #2:

Remarks to the Author:

Although clarifying the relevant stoichiometry(s) of Get1/Get2 in cells would strengthen the manuscript, the author's interpretation of their structures in light of previously published work supporting a heterotetramer is not unreasonable. Nevertheless, the main concerns of both reviewers remain after revision: namely, that the manuscript is a collection of preliminary observations that do not shed much new light on the mechanism of TA insertion. Unfortunately no new experiments or analysis is provided by the authors in this revision. We are left with observations that are no longer surprising (e.g., that the Get1/Get2 heterodimer structure is conserved across eukaryotes; that the membrane is thinned near Get1/Get2), or not developed (e.g., what is the relevance of the state 1 and state 2 conformations observed in different structures)? The impact of this manuscript remains quite limited and better suited to a specialty journal rather than Nature Communications.

Reviewer #3:

Remarks to the Author:

The GET complex of the ER membrane mediates the insertion of tail-anchored proteins. The machineries for the insertion of tail-anchored proteins into the ER membrane, the GET and the EMC complex, were identified only rather recently and the details of how these structures mediate protein insertion are still largely unresolved. This study by Sinning and colleagues presents detailed analyses from structural studies using cryo EM as well as from atomistic simulations and complementation experiments in baker's yeast. From these data the authors can draw several important conclusions: (1) By solving the structure of the Chaetomium complex, they provide evidence that the overall structure of the Get1-Get2 heterodimer is conserved between fungi and animals and that the EMC complex in different eukaryotic groups works by one uniform, common mechanism. (2) They show that helix $\alpha 3'$ of the Get2 subunit is a crucial regulatory element that drives conformational switching between different functional states of the Get1-Get2 complex. This switch element is shown here to be critical for a dynamic rearrangement in the Get1-Get2 heterotetramer that underlies the mechanism by which tail-anchored proteins are inserted. (3) The membrane-embedded module of the GET complex consists of a heterotetramer with two Get1 and two Get2 subunits. (4) The Get1-Get2 complex mediates protein insertion by membrane thinning and local distortion of membrane lipids, confirming previous models.

The study is of very high quality and shows for the first time the structure of the thermophilic fungus Chaetomium as well as that of GET complexes lacking the $\alpha 3'$ helix. This allows novel, exciting and important insights into the molecular mechanism by which the GET complex of fungi and animals mediate protein insertion. This is of outstanding relevance for a broad readership and represents – to my opinion – a considerable advance in the field.

Some parts of the study, in particular the first paragraphs of the results section, are difficult to read. The authors already tried to make their study better accessible during the revision, but readers still need to invest quite some effort to read and digest the study. However, this is also due to the fact that the structural details discussed cannot be easily presented in a "light" and easily accessible way. In summary, this is an exciting study of high quality and general interest. I support the publication of this study in its present form.

REVIEWERS' COMMENTS

Reviewer #1 (Remarks to the Author):

The presentation is much improved. The paper merits publication.

We are grateful for this clear and positive statement.

Reviewer #2 (Remarks to the Author):

Although clarifying the relevant stoichiometry(s) of Get1/Get2 in cells would strengthen the manuscript, the author's interpretation of their structures in light of previously published work supporting a heterotetramer is not unreasonable. Nevertheless, the main concerns of both reviewers remain after revision: namely, that the manuscript is a collection of preliminary observations that do not shed much new light on the mechanism of TA insertion. Unfortunately no new experiments or analysis is provided by the authors in this revision. We are left with observations that are no longer surprising (e.g., that the Get1/Get2 heterodimer structure is conserved across eukaryotes; that the membrane is thinned near Get1/Get2), or not developed (e.g., what is the relevance of the state 1 and state 2 conformations observed in different structures)? The impact of this manuscript remains quite limited and better suited to a specialty journal rather than Nature Communications.

We thank this reviewer for the time to read and comment on our manuscript. We appreciate that this reviewer finds our previous data defining the stoichiometry of the GET insertase reasonable. We regret if we could not convince referee 2 with the data provided in our manuscript and the arguments and information supplied also with the previous revision.

The current understanding of TA insertion is far from being complete. Our human structure concerns the post-insertion state, and in order to really understand the mechanism all individual states of TA delivery and insertion would be needed, which we cannot supply here. However, we disagree with the generic statement that our data "do not shed much new light on the TA insertion mechanism". The manuscript arrives at three main and important conclusions, which for this referee might not be surprising, but which are based on data that have not been reported before:

- *the conservation of the Get1/Get2 heterodimer across eukaryotes,*
- *the central importance of helix $\alpha 3'$ for the conformation of the insertase structure as a whole as well as on the Get3 TA binding domain,*
- *membrane thinning by the insertase.*

All three are of direct relevance for TA protein release and insertion.

As discussed in the manuscript, we are puzzled by the different conformations of the human and Chaetomium structures, which show that the insertase is much more dynamic than expected. While we cannot give a convincing trajectory between the two conformations (states 1 and 2), our study defines helix $\alpha 3'$ as an essential element for productive TA insertion with a direct impact on insertase structure. Moreover, the atomistic description of the mechanisms of membrane remodeling via thinning and bending, is just beginning, and

presents an exciting research area in membrane protein biochemistry. Overall, we are sure our manuscript will inspire the field and create an according impact.

Reviewer #3 (Remarks to the Author):

The GET complex of the ER membrane mediates the insertion of tail-anchored proteins. The machineries for the insertion of tail-anchored proteins into the ER membrane, the GET and the EMC complex, were identified only rather recently and the details of how these structures mediate protein insertion are still largely unresolved. This study by Sinning and colleagues presents detailed analyses from structural studies using cryo EM as well as from atomistic simulations and complementation experiments in baker's yeast. From these data the authors can draw several important conclusions: (1) By solving the structure of the Chaetomium complex, they provide evidence that the overall structure of the Get1-Get2 heterodimer is conserved between fungi and animals and that the EMC complex in different eukaryotic groups works by one uniform, common mechanism. (2) They show that helix a3' of the Get2 subunit is a crucial regulatory element that drives conformational switching between different functional states of the Get1-Get2 complex. This switch element is shown here to be critical for a dynamic rearrangement in the Get1-Get2 heterotetramer that underlies the mechanism by which tail-anchored proteins are inserted. (3) The membrane-embedded module of the GET complex consists of a heterotetramer with two Get1 and two Get2 subunits. (4) The Get1-Get2 complex mediates protein insertion by membrane thinning and local distortion of membrane lipids, confirming previous models.

The study is of very high quality and shows for the first time the structure of the thermophilic fungus Chaetomium as well as that of GET complexes lacking the a3' helix. This allows novel, exciting and important insights into the molecular mechanism by which the GET complex of fungi and animals mediate protein insertion. This is of outstanding relevance for a broad readership and represents – to my opinion - a considerable advance in the field. Some parts of the study, in particular the first paragraphs of the results section, are difficult to read. The authors already tried to make their study better accessible during the revision, but readers still need to invest quite some effort to read and digest the study. However, this is also due to the fact that the structural details discussed cannot be easily presented in a “light” and easily accessible way.

In summary, this is an exciting study of high quality and general interest. I support the publication of this study in its present form.

We thank this referee for this positive comment and the excitement about our data. We agree that some parts the story are quite complex and we apologize if this challenges the reader. However, we are happy to hear that this referee appreciates the structural details. We are grateful for the strong support for our manuscript in the current form.